# GRAPH STRUCTURE AND FEATURE EXTRAPOLATION FOR OUT-OF-DISTRIBUTION GENERALIZATION

## ABSTRACT

Out-of-distribution (OOD) generalization deals with the prevalent learning scenario where test distribution shifts from training distribution. With rising application demands and inherent complexity, graph OOD problems call for specialized solutions. While data-centric methods exhibit performance enhancements on many generic machine learning tasks, there is a notable absence of data augmentation methods tailored for graph OOD generalization. In this work, we propose to achieve graph OOD generalization with the novel design of non-Euclidean-space linear extrapolation. The proposed augmentation strategy extrapolates both structure and feature spaces to generate OOD graph data. Our design tailors OOD samples for specific shifts without corrupting underlying causal mechanisms. Theoretical analysis and empirical results evidence the effectiveness of our method in solving target shifts, showing substantial and constant improvements across various graph OOD tasks.

## 1 INTRODUCTION

Machine learning algorithms typically assume training and test data are independently and identically distributed (i.i.d.). However, distribution shift is a common problem in real-world applications, which substantially degrades model performance. The out-of-distribution (OOD) generalization problem deals with learning scenarios where test distributions shift from training distributions and remain unknown during the training phase. The area of OOD generalization has gained increasing interest over the years, and multiple OOD methods have been proposed (Ganin et al., 2016; Arjovsky et al., 2019; Krueger et al., 2021). Although both general OOD problems and graph analysis (Veličković et al., 2018; Liu et al., 2021b; Gui et al., 2022b) have been intensively studied, graph OOD research is only in its early stage (Wu et al., 2022b;a; Bevilacqua et al., 2021; Gui et al., 2023). With various applications and unique complexity, graph OOD problems call for specific solutions. Data augmentation (DA) methods have shown a significant boost in generalization capability and performance improvement across multiple fields (Shorten and Khoshgoftaar, 2019; Yao et al., 2022; Park et al., 2022; Han et al., 2022), creating a promising possibility for graph OOD studies. Currently, environment-aware DA methods for graph OOD are under-explored.

Conventional data augmentations increase the amount of data and act as regularizers to reduce over-fitting, which empirically enhances model performance in previous studies (Rong et al., 2019; You et al., 2020). Many DA techniques (Zhang et al., 2017; Yao et al., 2022; Wang et al., 2021), including graph data augmentations (GDA), exclusively interpolate data samples to generate new ones. Interpolation-based DA boosts model performances by making overall progress in learning (Xu et al., 2020), Mixup (Zhang et al., 2017) being a typical example. However, practical tasks are often out-of-distribution instead of in-distribution (ID). Thus, models are expected to extrapolate instead of interpolate. Currently, few augmentation studies focus on extrapolation, especially for graphs. The distribution area where models cannot generalize to is also hardly reachable when generating augmentation samples using traditional techniques, which is a substantial obstacle to OOD generalization. Although graphon calculation (Han et al., 2022) is a potential avenue to extrapolate, the lack of consideration in environment information and causal design renders its causal correlations easily breached. Thus, the performance gain from DA appears limited in OOD tasks.

In this work, we propose to solve OOD generalization in graph classification tasks from a data-centric perspective. To stimulate the potential improvement of DA in OOD tasks, we study graph-space extrapolation, essentially, generating OOD data samples. Graph data has the complex nature of topological irregularity and connectivity, with unique types of distribution shifts in both feature and structure. Theoretically, it is impossible to solve unknown shifts without auxiliary information (Lin et al., 2022). Thus, injecting environment information (Gulrajani and Lopez-Paz, 2020) in training to convey the types of shifts is a necessary and promising solution for OOD. We innovatively propose an

*environment-aware* framework with non-Euclidean-space linear extrapolation designed in both graph structural and feature space. Our contributions are three-fold. Firstly, we establish non-Euclidean space linear extrapolation with definitions, analyses, and guarantees. We theoretically justify that samples generated from linear extrapolation follow common causal assumptions and are tailored for specific OOD shifts. Secondly, we instantiate graph extrapolation as an efficacious generalization method. Structural linear extrapolation is enabled with novel graph splicing and label-environment-aware pair learning techniques, while feature linear extrapolation performs space-spanning with causally selected features. To the best of our knowledge, we are the first GDA method to achieve both feature and structural extrapolation, and further cover complete structural global/local extrapolation in both distribution directions. Thirdly, our extensive experiments validate the superiority of our method over both graph invariant learning and data augmentation baselines for complex shifts.

**Comparison with prior methods.** While previous graph OOD methods (Wu et al., 2022b; Miao et al., 2022; Li et al., 2022; Chen et al., 2022b) focus on invariant learning regularization, we target OOD generalization from an explicit data-centric perspective. The major distinction between our method and the augmented-based learning strategy DIR (Wu et al., 2022b) is the ability to generate new graphs. Specifically, DIR mixes the embedding of subgraphs but is unable to generate new data in the input space, limiting its application scope of augmentation to single trainings. In contrast, our method aims to produce explict OOD graphs, which can be easily transferred and jointly used with other graph datasets. In the context of DA, current graph OOD augmentation methods either augment in embedding level (Kwon et al., 2022) or limit the augmented data to subgraphs of the original data (Yu et al., 2022; Sui et al., 2022). On the contrary, we define innovative extrapolation operators in the input level rigorously and generate diverse unseen graphs of functional sizes, backbones, and features. A more detailed discussion of related works is provided in Appendix A.

## 2 PROBLEM SETTING

**Graph notations.** We denote a graph as $G = (\boldsymbol{A}, \boldsymbol{X}, \boldsymbol{E})$, where $\boldsymbol{A} \in \mathbb{R}^{n \times n}$, $\boldsymbol{X} \in \mathbb{R}^{p \times n}$, and $\boldsymbol{E} \in \mathbb{R}^{q \times m}$ are the adjacency, node feature, and edge feature matrices, respectively. We assume $n$, $m$, $p$, and $q$ are the numbers of nodes, edges, node features, and edge features respectively. Additionally, we assume a set of latent variables $\{\boldsymbol{z}_i \in \mathbb{R}^f\}_{i=1}^n$ form a matrix $\boldsymbol{Z} = [\boldsymbol{z}_1, \boldsymbol{z}_2, \cdots, \boldsymbol{z}_n] \in \mathbb{R}^{f \times n}$. For graph-level tasks, each graph has a label $Y \in \mathcal{Y}$, and for node-level tasks there is a label per node.

**OOD settings.** The environment formalism following invariant risk minimization (IRM) is a common setting for OOD studies (Peters et al., 2016; Arjovsky et al., 2019; Krueger et al., 2021). This framework assumes that training data form groups, known as environments. Data are similar within the same environment but dissimilar across different environments. Since multiple shifts can exist between training and test data, models are usually not expected to solve all shifts. Instead, the target shift type is conveyed using environments. Specifically, the target shift between training and test data, though more significant, should be similarly reflected among different training environments. In this case, OOD methods can potentially grasp the shift by learning among different environments. In this work, we follow the formalism and use environment information in augmentation strategies to benefit OOD generalization. Environments are given as environment labels $\varepsilon_i \in \mathcal{E}$ for each data sample.

**Graph structure and feature distribution shifts.** Graph data contains complex topologies. Graph distribution shifts can happen on both features and structures, which possesses different properties and should be handled separately. Feature distribution shifts happen on node or edge features, and we consider node features in this work. In this case, shifts are solely on the node feature distribution as $P^{\text{tr}}(\boldsymbol{X}) \neq P^{\text{te}}(\boldsymbol{X})$, while $P^{\text{tr}}(\boldsymbol{A}) = P^{\text{te}}(\boldsymbol{A})$, where $P^{\text{tr}}(\cdot)$ and $P^{\text{te}}(\cdot)$ denote training and test distributions, respectively. In contrast, structure distribution shift is the more distinctive and complex case in graph OOD. Structural shifts can happen in the distribution of $\boldsymbol{A}$ or the conditional distribution between $\boldsymbol{X}$ and $\boldsymbol{A}$, resulting in $P^{\text{train}}(\boldsymbol{X}, \boldsymbol{A}) \neq P^{\text{test}}(\boldsymbol{X}, \boldsymbol{A})$. In graph-level tasks, structural shifts exist both globally and locally. Common global/local domain examples are graph size and graph base (Hu et al., 2020; Gui et al., 2022a), the latter also known as scaffold (Bemis and Murcko, 1996) in molecule data. Specifically, graph size refers to its number of nodes, and graph base refers to the non-functional backbone substructure irrelevant with targets.

## 3 LINEAR EXTRAPOLATION IN GRAPH SPACE

We propose the GDA strategy of input-space linear extrapolation, inspired by the philosophy of "linear interpolation" in Mixup (Wang et al., 2021). Linear extrapolation of data distributions essentially

guides the model to behave outside the original range by introducing reachable OOD samples. In this section, we define linear extrapolation to extend beyond training distributions in both structure and feature spaces for graph data, effectively teaching the model to anticipate and handle OOD scenarios.

## 3.1 CAUSAL ANALYSIS

We first establish causal analyses following prior invariant learning works (Ahuja et al., 2021; Rosenfeld et al., 2020; Lu et al., 2021). As shown in Figure 1(b), $C, S_1, S_2 \in \mathcal{Z}$ are the latent variables in high-dimensional space that are causally associated with the target $Y$, non-causally associated with $Y$, and independent of $Y$, respectively. The environment $\mathcal{E}$ is target-irrelevant and observable. In the non-Euclidean space, we posit that information from the latent space is wholly reflected in the graph structure, so that $C$ and $\mathcal{E}$ determine respective subgraphs of a graph (Chen et al., 2022b;c). Formally, we define subgraphs caused by $C$ as causal subgraphs $G_{\text{inv}}$, and subgraphs caused by $\mathcal{E}$ as environmental subgraphs $G_{\text{env}}$. Since graphs with the same label should contain invariant causal subgraphs, causal subgraphs are potentially extractable from label-invariant graphs, and environmental subgraphs from environment-invariant graphs. Considering distribution shifts in the feature space, we assume $C$ and $\mathcal{E}$ determine respective elements of feature vectors. For single-node feature $\boldsymbol{x} \in \mathbb{R}^p$, where $\boldsymbol{x} = \boldsymbol{X}$ for node-level tasks and $\boldsymbol{x} \subseteq \boldsymbol{X}$ for graph-level tasks, let $p = i + v$. We define node features determined by $C$ as invariant node features $\boldsymbol{x}_{\text{inv}} \in \mathbb{R}^i$, while other node features are variant features $\boldsymbol{x}_{\text{var*}} \in \mathbb{R}^v$. In practice, with environment information, it is realistic to assume we can learn to select a subset of the variant features $\boldsymbol{x}_{\text{var}} \in \mathbb{R}^j$ substantially determined by $\mathcal{E}$, where $j \leq v$.

## 3.2 LINEAR EXTRAPOLATION FORMULATION

Linear extrapolation, which constructs samples beyond the known range while maintaining the same direction and magnitude of known sample differences, is a central concept in our approach.

**Definition 1.** [Feature Linear Extrapolation (FLE)] Given two feature vector points $(\boldsymbol{x}_i, y_i)$ and $(\boldsymbol{x}_j, y_j)$, feature linear extrapolation is defined as

$$\boldsymbol{x}_{fle} = \boldsymbol{x}_i + a(\boldsymbol{x}_j - \boldsymbol{x}_i), y_{fle} = y_i + a(y_j - y_i), \quad s.t. \quad a \in \mathbb{R}, a > 1 \vee a < 0. \tag{1}$$

The extension of linear extrapolation to graph structure requires the definition of structural linear calculations. We define graph addition, $G_1 + G_2$, as the splicing of two graphs, resulting in unions of their vertex and edge sets. Graph subtraction, $G_2 - G_1$, is defined as subtracting the largest isomorphic subgraph of $G_1$ and $G_2$ from $G_2$. Let $D\{G_{tr}\} = \{(G_1, y_1), (G_2, y_2), \ldots, (G_N, y_N)\}$ be the $N$-sample graph training set. Given the discrete nature of graph operations, we formulate the linear extrapolation of graph structures below.

**Definition 2.** [Structural Linear Extrapolation (SLE)] Given graphs $G_i, G_j \in D\{G_{tr}\}$, we define 1-dimension structural linear extrapolation on $D\{G_{tr}\}$ as $G_{sle}^1 = a_i \cdot G_i + b_{ij} \cdot (G_j - G_i)$, where $a_i, b_{ij} \in \{0, 1\}$. We extend to define N-dimension structural linear extrapolation:

$$G_{sle}^N = \sum_{i=1}^N a_i \cdot G_i + \sum_{i=1}^N \sum_{j=1}^N b_{ij} \cdot (G_j - G_i) = \mathbf{a}^\top \mathbf{G} + \langle B, \mathbf{1}\mathbf{G}^\top - \mathbf{G}\mathbf{1}^\top \rangle_F, \tag{2}$$

where $\mathbf{a} = [a_1, \ldots, a_N]^\top$, $B = \{b_{ij}\}^{N \times N}$, $\mathbf{G} = [G_1, \ldots, G_N]^\top$, $\mathbf{1}$ is a $N$-element vector of ones, and $\langle \cdot, \cdot \rangle_F$ is the Frobenius inner product. Let $c_{ij} \in \{0, 1\}$ indicate the existence of causal subgraphs in $(G_j - G_i)$. Then the label for $G_{sle}^N$ is defined as $y_{sle}^N = (\sum_{i=1}^N a_i \cdot y_i + \sum_{i=1}^N \sum_{j=1}^N c_{ij}b_{ij} \cdot y_j)/(\sum_{i=1}^N a_i + \sum_{i=1}^N \sum_{j=1}^N c_{ij}b_{ij}) = (\mathbf{a}^\top \mathbf{y} + \langle C \circ B, \mathbf{1}\mathbf{y}^\top \rangle_F)/(\mathbf{a}^\top \mathbf{1} + \langle C, B \rangle_F)$, where $\circ$ denotes Hadamard product, $\mathbf{y} = [y_1, \ldots, y_N]^\top$, and $C = \{c_{ij}\}^{N \times N}$.

Note that we do not need to avoid multiple graphs to ensure linearity in Eq. 2, due to the high dimensionality of graph structure. In this context, $\mathbf{a}^\top \mathbf{G}$ denotes splicing multiple graphs together; while $\langle B, \mathbf{1}\mathbf{G}^\top - \mathbf{G}\mathbf{1}^\top \rangle_F$ denotes splicing together multiple subtracted subgraphs. These definitions enable structural linear extrapolation in the non-Euclidean graph space.

## 3.3 LINEAR EXTRAPOLATION FOR OOD GENERALIZATION

In this subsection, we justify that linear extrapolation can generate OOD samples respecting specific shifts while maintaining causal validity, i.e., preserving underlying causal mechanisms. First, we establish an assumption that combining causal structures causes a sample with mixed label for SLE.

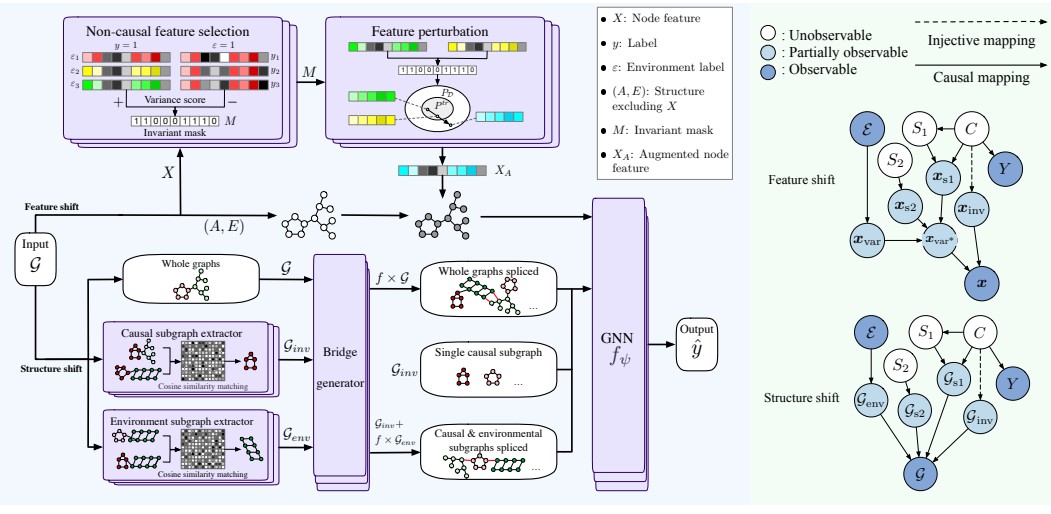

(a) Architecture Overview        (b) Causal Graphs

Figure 1: (a) Architecture overview. Causal features are selected and then preserved along with graph topological structure, while extrapolation on non-causal features spans their feature space to solve feature shifts. For structure shifts, diverse component graphs are extracted with pair-learning approaches, selected and spliced with extrapolation strategies to achieve causally-valid structural OOD samples. (b) Illustration of causal graphs. Unobservable variables lie in the latent space; partially observable variables can be learned and selected.

**Assumption 1** (Causal Additivity). *Let $(G_1, y_1)$, $(G_2, y_2)$ and $(G_3, y_3)$ be graph-label pairs. If $G_3 = G_{1inv} + G_{2inv} + G_{env}$, then $y_3 = ay_1 + (1-a)y_2$, where $G_{env}$ is any combination of environmental subgraphs, and $a \in (0, 1)$.*

This causal assumption holds valid in a wide range of graph classification tasks, and we discuss its scope of application in Appendix D. Next, we provide two definitions to establish the conditions under which structural linear extrapolation can cover certain global and local environment values.

**Definition 3.** [Global SLE Achievability] Given a $N$-sample set of graphs $D\{G\}$, we say that a graph global property $gp(\boldsymbol{A}, \boldsymbol{X}, \boldsymbol{E})$ is achievable by global SLE if there exists an N-dimension structural linear extrapolation $G_{sle}^N$ on $D\{G\}$, s.t. $G_{sle}^N = (\boldsymbol{A}, \boldsymbol{X}, \boldsymbol{E})$.

**Definition 4.** [Local SLE Achievability] Given a $N$-sample set of graphs $D\{G\}$, we say that a substructure $\mathcal{B}$ is achievable by local SLE if there exists an N-dimension structural linear extrapolation $G_{sle}^N$ on $D\{G\}$, s.t. $(G_{sle}^N)_{env} = \mathcal{B}$.

The following theorems assert that structural linear extrapolation can create OOD samples covering at least two environments in opposite directions of the distribution, respecting global/local shifts in size and base each, and ensure the causal validity of these samples.

**Theorem 3.1.** *Given an $N$-sample training dataset $D\{G_{tr}\}$, its N-dimension structural linear extrapolation can generate sets $D\{G_1\}$ and $D\{G_2\}$ s.t. $(G_1)_{env} < (G_{tr})_{env} < (G_2)_{env}$ for $\forall G_{tr}, G_1, G_2$, where $<$ denotes "smaller in size" and "lower base complexity" for size and base extrapolation.*

**Theorem 3.2.** *Given an $N$-sample training dataset $D\{G_{tr}\}$ and its true labeling function for the target classification task $f(G)$, if $D\{G_{sle}^N\}$ is a graph set sampled from N-dimension structural linear extrapolation of $D\{G_{tr}\}$ and Assumption 1 holds, then for $\forall(G_{sle}^N, y) \in D\{G_{sle}^N\}$, $y = f(G_{sle}^N)$.*

Proofs and further analysis are in Appendix G. These theorems show that structural linear extrapolation has the capability to generate OOD samples that are both plausible and diverse. The justification of feature linear extrapolation is relatively straightforward and provided in Section 5.3. Thus far, we have provided theoretical bases for the applicability of linear extrapolation on graph OOD tasks.

## 4   G-SPLICE FOR STRUCTURAL LINEAR EXTRAPOLATION

In this section, we specify structural linear extrapolation as a feasible augmentation method with detailed implementations, termed G-Splice. Using environment information, the method extrapolates global structural features while preserving structural information that causes the label. The approach is underpinned by theoretical analysis for structural linear extrapolation, providing causally-valid

OOD samples that are tailored for specific shifts. The overall model constructs diverse augmentation samples, as shown in Figure 1(a). In the following subsections, we describe the technical modules of splicing, component graph selection, and post-sampling procedures separately.

## 4.1 GRAPH SPLICE

The action of splicing a group of component graphs is essentially a conditional edge generation task, which we refer to as "bridge" generation. We generate bridges of predicted number along with corresponding edge attributes between given component graphs to join multiple components into a single graph. In this work, we use conditional variational autoencoders (cVAE) (Kingma and Welling, 2013; Kipf and Welling, 2016), though other generative models may also be used. We adopt cVAE as the major bridge generator for its adequate capability and high efficiency, as compared with diffusion models (Ho et al., 2020) in Appendix I. The bridge generator takes as input a group of component graphs, denoted as $G_1, \cdots, G_f = (\boldsymbol{X}_1, \boldsymbol{A}_1, \boldsymbol{E}_1), \cdots, (\boldsymbol{X}_f, \boldsymbol{A}_f, \boldsymbol{E}_f)$. The cVAE encoder produces a latent variable distribution. Specifically, we construct the inference model as

$$q_\phi(\boldsymbol{Z}|G_1, \cdots, G_f) = \prod_{\substack{i=1 \\ v_i \sim G_j}}^{n} q_\phi(\boldsymbol{z}_i|\boldsymbol{X}_j, \boldsymbol{A}_j, \boldsymbol{E}_j) = \prod_{i=1}^{n} \mathcal{N}(\boldsymbol{z}_i|\boldsymbol{\mu}_i, \mathrm{diag}(\boldsymbol{\sigma}_i^2)), \quad (3)$$

where $v_i \sim G_j$ denotes that the $i$-th node $v_i$ belongs to component graph $G_j$, $\boldsymbol{\mu}_i$ and $\boldsymbol{\sigma}_i$ are the generated mean and standard deviation vectors of the $i$-th latent distribution, and $n$ is the total number of nodes in all component graphs. The encoder $q_\phi$ is parameterized by three-layer graph isomorphism networks (GIN) (Xu et al., 2019a). The generative model produces the probability distribution for bridges $\boldsymbol{A}^b$ and corresponding attributes $\boldsymbol{E}^b$:

$$p_\theta(\boldsymbol{A}^b, \boldsymbol{E}^b|\boldsymbol{Z}) = \prod_{i=1}^{n} \prod_{j=1}^{n} p_\theta(\boldsymbol{A}_{ij}^b, \boldsymbol{e}_{ij}^b|\boldsymbol{z}_i, \boldsymbol{z}_j),$$

$$p_\theta(\boldsymbol{A}_{ij}^b, \boldsymbol{e}_{ij}^b|\boldsymbol{z}_i, \boldsymbol{z}_j) = \begin{cases} \mathrm{MLP}_\theta(\boldsymbol{z}_i, \boldsymbol{z}_j), & \text{if } v_i \sim G_s, v_j \sim G_t, \\ & \qquad\quad s.t. \quad s \neq t \\ (0, \mathrm{None}), & \text{otherwise} \end{cases}$$

where $\boldsymbol{A}_{ij}^b$ is the $ij$-th element of $\boldsymbol{A}^b$ and $\boldsymbol{e}_{ij}^b \in \boldsymbol{E}^b$ is the corresponding edge attribute vector. By sampling $B$ times from $p_\theta(\boldsymbol{A}^b, \boldsymbol{E}^b|\boldsymbol{Z})$, we sample $B$ pairs of bridge-attribute vectors to complete bridge generation. To train the bridge generator, we optimize the variational lower bound $\mathcal{L}$ w.r.t. the variational parameters by $\mathcal{L}_{\theta,\phi} = \mathbb{E}_{q_\phi} \log p_\theta(\boldsymbol{A}^b|\boldsymbol{Z}) + \alpha \mathbb{E}_{q_\phi} \log p_\theta(\boldsymbol{E}^b|\boldsymbol{Z}) - \beta \mathrm{KL}[q_\phi||p(\boldsymbol{Z})]$, where $\mathrm{KL}[q(\cdot)||p(\cdot)]$ is the Kullback-Leibler divergence between $q(\cdot)$ and $p(\cdot)$. We take the Gaussian prior $p(\boldsymbol{Z}) = \prod_i p(\boldsymbol{z}_i) = \prod_i \mathcal{N}(\boldsymbol{z}_i|0, \boldsymbol{I})$. $\alpha$ and $\beta$ are hyperparameters regularizing bridge attribute and KL divergence respectively. Note that we do not include new nodes as part of the bridge, since we aim at preserving the local structures of the component graphs and extrapolating certain global features. More manually add-on graph structures provide no extrapolation significance, while their interpolation influence are not proven beneficial, which is reflected in Appendix I.

**Bridge number prediction.** To predict the number of prospective bridges between a set of component graphs, a pre-trained GNN parameterized by $\eta$ produces probabilities for the bridge number $B$, $p_\eta(B) = \mathrm{GNN}_\eta(\boldsymbol{X}_1, \boldsymbol{A}_1, \boldsymbol{E}_1, \cdots, \boldsymbol{X}_f, \boldsymbol{A}_f, \boldsymbol{E}_f)$. When generating bridges, we first sample the number $B$ with the predicted probabilities from the categorical distribution.

## 4.2 COMPONENT GRAPH SELECTION

**Whole graphs.** Corresponding to $\mathbf{a}^\top \mathbf{G}$ in Eq. 2, we use whole graphs from the training data as a category of component graphs, which possesses computational simplicity and enables extrapolation.

**Causal subgraphs and environmental subgraphs.** The part of $\langle B, \mathbf{1}\mathbf{G}^\top - \mathbf{G}\mathbf{1}^\top \rangle_F$ in Eq. 2 requires the operation of subtracting the largest isomorphic subgraph, which is practically unfeasible. In this case, using target and environment label information, we approximate $(G_j - G_i)$ for particular graph pairs. Specifically, $(G_j - G_i) \approx G_{j\mathrm{inv}}$ for $G_j, G_i$ with different labels but the same environment, and $(G_j - G_i) \approx G_{j\mathrm{env}}$ for $G_j, G_i$ with the same label but different environments. Therefore, we can use extracted causal/environmental subgraphs as $(G_j - G_i)$. We perform pair-wise similarity matching on label-invariant graphs to extract causal subgraphs, and on environment-invariant graphs to extract environmental subgraphs, following Sec. 3.1. Since environment for test data remains unknown and these subgraphs are inaccessible during test, using these subgraphs in data augmentation for

training can be an optimum strategy. Let $G$ and $G'$ be two graphs with the same label $y_1$ but different environments $\varepsilon_1$ and $\varepsilon_2$. We identify $G_{inv}$ as the subgraph shared by both graphs that exhibits the highest similarity. The theorem below establishes that this identification approach unveils the causal subgraph $G_{inv}$ by definition under certain causal assumptions.

**Theorem 4.1.** *Given the causal graph (Figure 1(b)) and assuming a bijective causal mapping between $C$ and $Y$, for $G$ and $G'$, let $G_s$ represent potential subgraphs. If $I(\cdot)$ measures cosine similarity in the graph embedding space and $f_z : \mathcal{G} \to \mathbb{R}^f$ is a feature mapping that reversely infers hidden features, then: (1) It follows that the similarity of the invariant subgraphs of $G$ and $G'$, $I\big(f_z(G_{inv}), f_z(G'_{inv})\big)$, reaches its maximal 1; (2) Given a subgraph set $\mathbf{G}_s = \{G_s | \exists G'_s s.t. I(f_z(G_s), f_z(G'_s)) = 1\}$, the invariant subgraph $G_{inv}$ can be obtained as $G_{inv} = \arg\max_{G_s \in \mathbf{G}_s} |G_s|$.*

Discussions and proofs are provided in Appendix G. Formally, the algorithmic procedure to identify $G_{inv}$ has optimization objective $G_{inv} = \arg\max_{G_s \in \mathbf{G}_s} |G_s|$, where the set of subgraphs $\mathbf{G}_s = \big\{G_s | \arg\max_{G_s} I(f(G_s), f(G'_s))\big\}$. Note that considering the injected noises of $G_{inv}$ in real world, though generally much smaller than $S_1$ and $S_2$, we adopt a relaxed version of $\mathbf{G}_s$. We use label-invariant and environment-variant graph pairs to pre-train a causal subgraph extraction network, which is optimized by the sampled causal subgraphs predicting the label $Y$ solely. Therefore, the outer training objective of $f_z$ is $f_z^* = \arg\min_{f_z} \ell(y; f_c(f_z(G_{inv})))$, where $f_c$ is the classifier. More algorithmic details are in Appendix C. Similarly, environment-invariant and label-variant graph pairs are used to pre-train the environmental subgraph extraction network using the environment label.

### 4.3 POST-SAMPLING PROCEDURE

To enable structural linear extrapolation, we need to make selections of component graphs and sample bridges accordingly, as well as assign the labels and environments. Since linear extrapolation has infinite choices of $a$ and $b$, to enable training, we simplify the augmentation into three options: 1.single causal subgraphs $G_{inv}$, 2.causal and environmental subgraphs spliced $G_{inv} + f \cdot G_{env}$, 3.whole graphs spliced $f \cdot G$. With the pre-trained component graph extractors and bridge generator, we apply these strategies to augment graph data in OOD classification tasks. The actual number of component graphs $f$ is set as a hyperparameter, tuned and determined through OOD validation during training. The augmentation selections are also tuned as a hyperparameter, with at least one option applied. We label the generated graphs following the definition of structural linear extrapolation. Considering extrapolation in environments, for size OOD tasks, we create up to three new environments with the three options according to the size distribution of the augmented graphs. For base/scaffold OOD tasks, we create up to two new environments, one for $G_{inv}$, the other for $G_{inv} + f \cdot G_{env}$ and $f \cdot G$, since the former option construct graphs without base/scaffolds, and the latter options graphs with multiple base/scaffolds combined. All original graphs and augmentation graphs then form the augmented training distribution with add-on environments. To make adequate use of the augmented environment information for OOD generalization, we optionally apply an invariant regularization (Krueger et al., 2021) during learning, reaching our final objective,

$$\psi^* := \arg\min_{\psi} \mathbb{E}_{(G,y) \sim \cup_{\varepsilon \in \{\mathcal{E} \cup \mathcal{E}_A\}} P_\varepsilon} [\ell(y; f_\psi(G))] + \gamma \mathrm{Var}_{\varepsilon \in \{\mathcal{E} \cup \mathcal{E}_A\}} [\mathbb{E}_{(G,y) \sim P_\varepsilon} \ell(y; f_\psi(G))], \quad (4)$$

where $\mathcal{E}_A$ are the augmented environments, $f_\psi$ is the prediction network for OOD tasks, $\ell(\cdot)$ calculates cross-entropy loss, and $\mathrm{Var}[\cdot]$ calculates variance.

## 5 FEATX FOR FEATURE LINEAR EXTRAPOLATION

We implement feature linear extrapolation as FeatX, a simple label-invariant data augmentation strategy designed to improve OOD generalization for graph feature shifts, which is applicable to both graph-level and node-level tasks, as shown in Figure 1. Environment information is used to selectively perturb non-causal features while causal features determining the label are preserved. During the augmentation process, with knowledge of the domain range of the node features, we span the feature space with extrapolation for non-causal features. In contrast to Mixup which exclusively interpolates, with knowledge of non-causal features and their domain, FeatX covers extrapolation as well as interpolation to advance in OOD generalization. We introduce technical details in the following subsections and end with theoretical support for the effectiveness of feature linear extrapolation.

### 5.1 NON-CAUSAL FEATURE SELECTION

Unlike graph structure, features reside in a relatively low-dimensional space. Direct extrapolation of node features that have causal relationships with the target may not yield beneficial outcomes

(Appendix I). Therefore, we only perturb the selected variant features $\boldsymbol{x}_{\text{var}}$. The selection of $\boldsymbol{x}_{\text{var}}$ is implemented as learning an invariance mask $\boldsymbol{M} \in \mathbb{R}^p$ based on variance score vector $\boldsymbol{S_V} \in \mathbb{R}^p$ and threshold $T$. The variance score vector $\boldsymbol{S_V}$ measures the variance of each feature element w.r.t. the target; high scores represents large variances and therefore features of $\boldsymbol{x}_{\text{var}}$. Variance scores are learned using label and environment information. Label-invariant and environment-variant samples should have similar invariant features $\boldsymbol{x}_{\text{inv}}$ while variant features $\boldsymbol{x}_{\text{var}}$ vary majorly; thus their feature variances increase the variance score vector $\boldsymbol{S_V}$. Conversely, feature variances of environment-invariant label-variant samples reduce variance scores. The invariance mask $\boldsymbol{M}$ selects $\boldsymbol{x}_{\text{var}}$ and masks out other features by applying threshold $T$ on variance score vector $\boldsymbol{S_V}$. Formally, $\boldsymbol{S_V} = k_1 \mathbb{E}_{y \in Y} \text{Var}_{P_y}[\boldsymbol{x}] - k_2 \mathbb{E}_{\varepsilon \in \mathcal{E}} \text{Var}_{P_\varepsilon}[\boldsymbol{x}], \boldsymbol{M} = [\boldsymbol{S_V} > T]$, where $\boldsymbol{k} = [k_1, k_2]$ and $T$ are trainable parameters, and $\text{Var}_P[\boldsymbol{x}]$ calculates variance of node features for samples in distribution $P$.

## 5.2 Node Feature Extrapolation

We apply the mask $\boldsymbol{M}$ and perturb the non-causal node features to achieve extrapolation w.r.t. $\boldsymbol{x}_{\text{var}}$, without altering the topological structure of the graph. Let the domain for $\boldsymbol{x}$ be denoted as $\mathcal{D}$, which is assumed to be accessible. Valid extrapolations must generate augmented samples with node feature $\boldsymbol{X}_A \in \mathcal{D}$ while $\boldsymbol{X}_A \nsim P^{\text{train}(\boldsymbol{X})}$. We ensure the validity of extrapolation with the generalized modulo operation (Appendix G), $\forall \boldsymbol{X} \in \mathbb{R}^p$, $(\boldsymbol{X} \bmod \mathcal{D}) \in \mathcal{D}$. Given each pair of samples $\boldsymbol{D}_{\varepsilon_1}, \boldsymbol{D}_{\varepsilon_2}$ with the same label $y$ but different environments $\varepsilon_1$ and $\varepsilon_2$, FeatX produces $\boldsymbol{X}_A = \boldsymbol{M} \times ((1+\lambda)\boldsymbol{X}_{\varepsilon_1} - \lambda' \boldsymbol{x}_{\varepsilon_2}) \bmod \mathcal{D} + \overline{\boldsymbol{M}} \times \boldsymbol{X}_{\varepsilon_1}, (\boldsymbol{A}, \boldsymbol{E}) = (\boldsymbol{A}_{\varepsilon_1}, \boldsymbol{E}_{\varepsilon_1})$, where $\lambda, \lambda' \sim \mathcal{N}(a, b)$ is sampled for each data pair. Empirically, we achieve favorable extrapolation performance and faster convergence with $\lambda' = \lambda \in \mathbb{R}^+ \sim \Gamma(a, b)$, which we use in experiments, with the shape parameter $a$ and scale parameter $b$ of gamma distribution as hyperparameters. Note that $\boldsymbol{D}_{\varepsilon_i}, \boldsymbol{D}_{\varepsilon_j}$ can be both graph-level and node-level data samples, and in the graph-level case $\boldsymbol{x}_{\varepsilon_2} \subseteq \boldsymbol{X}_{\varepsilon_2}$ is the features of a random node. The augmented samples form a new environment. Replacing original node features in training samples by the augmented ones, our optimization process is formulated as

$$\psi^* := \underset{\psi, \boldsymbol{k}, T}{\text{argmin}} \, \mathbb{E}_{(\boldsymbol{D}_{\varepsilon_i}, \boldsymbol{D}_{\varepsilon_j}, y) \sim P^{\text{train}}}[\ell(f_\psi(\boldsymbol{X}_A, \boldsymbol{A}, \boldsymbol{E}), y)],$$

where $f_\psi$ is the prediction network for OOD tasks and $\ell(\cdot)$ calculates cross-entropy loss.

## 5.3 Solving Feature-based Graph Distribution Shifts

FeatX enables extrapolation w.r.t. the selected variant features, while causal features are preserved, thereby transforming OOD areas to ID. Theoretical analysis can evidence that our extrapolation spans the feature space outside $P^{\text{train}}(\boldsymbol{X})$ for $\boldsymbol{x}_{\text{var}}$. We present the following theorem showing that, under certain conditions, FeatX substantially solves feature shifts on the selected variant features for node-level tasks. Let $n_A$ be the number of samples FeatX generates and $f_\psi$ be the well-trained network with FeatX applied.

**Theorem 5.1.** *If (1) $\exists (\boldsymbol{X}_1, \cdots, \boldsymbol{X}_j) \in P^{train}$ from at least 2 environments, s.t. $(\boldsymbol{X}_{1var}, \cdots, \boldsymbol{X}_{jvar})$ span $\mathbb{R}^j$, and (2) $\forall \boldsymbol{X}_1 \neq \boldsymbol{X}_2$, the GNN encoder of $f_\psi$ maps $G_1 = (\boldsymbol{X}_1, \boldsymbol{A}, \boldsymbol{E})$ and $G_2 = (\boldsymbol{X}_2, \boldsymbol{A}, \boldsymbol{E})$ to different embeddings, then with $\hat{y} = f_\psi(\boldsymbol{X}, \boldsymbol{A}, \boldsymbol{E})$, $\hat{y} \perp\!\!\!\perp \boldsymbol{X}_{var}$ as $n_A \to \infty$.*

Proof is provided in Appendix G. Theorem 5.1 states that, given sufficient diversity in environment information and expressiveness of GNN, FeatX can achieve invariant prediction regarding the selected variant features. Therefore, FeatX possesses the capability to generalize over distribution shifts on the selected variant features. Extending on the accuracy of non-causal selection, if $\boldsymbol{x}_{\text{var*}} = \boldsymbol{x}_{\text{var}}$, we achieve causally-invariant prediction in feature-based OOD tasks.

## 6 Experimental Studies

We evaluate the effectiveness of our method on multiple graph OOD classification tasks.

**Setup.** For all experiments, we select the best checkpoints for OOD tests according to results on OOD validation sets; ID validation and ID test are also used for comparison if available. For fair comparisons, we use unified GNN backbones for all methods in each dataset, specifically, GIN-Virtual (Xu et al., 2019a; Gilmer et al., 2017) and GCN (Kipf and Welling, 2017) for graph-level and node-level tasks, respectively. Experimental details and hyperparameters are provided in Appendix H.

Table 1: Performances of 16 baselines and G-Splice on 8 datasets with structure shift. ↑ indicates higher values correspond to better performance. $ID_{ID}$ denotes ID test results with ID validations, while $OOD_{OOD}$ denotes OOD test results with OOD validations. All numerical results are averages across 3+ random runs. Numbers in **bold** represent the best results and underline the second best.

| structure | GOOD-HIV-size↑ | | GOOD-HIV-scaffold↑ | | GOOD-SST2-length↑ | | Twitter-length↑ | | GOOD-Motif-size↑ | | GOOD-Motif-base↑ | | DD-size↑ | NCI1-size↑ |
|---|---|---|---|---|---|---|---|---|---|---|---|---|---|---|
| | $ID_{ID}$ | $OOD_{OOD}$ | $ID_{ID}$ | $OOD_{OOD}$ | $ID_{ID}$ | $OOD_{OOD}$ | $ID_{ID}$ | $OOD_{OOD}$ | $ID_{ID}$ | $OOD_{OOD}$ | $ID_{ID}$ | $OOD_{OOD}$ | $OOD_{OOD}$ | $OOD_{OOD}$ |
| ERM | 83.72±1.06 | 59.94±2.86 | 82.79±1.10 | 69.58±1.99 | 89.82±0.01 | 81.30±0.35 | **65.70±1.21** | 57.04±1.70 | 92.28±0.10 | 51.74±2.27 | 92.60±0.03 | 68.66±3.43 | 0.15±0.11 | 0.16±0.10 |
| IRM | 81.33±1.13 | 59.94±1.59 | 81.35±0.83 | 67.97±2.46 | 89.41±0.11 | 79.91±1.97 | 64.02±0.56 | 57.72±1.03 | 92.18±0.09 | 53.68±4.11 | 92.60±0.02 | 70.65±3.18 | 0.06±0.08 | 0.11±0.07 |
| VREx | 83.47±1.11 | 58.49±2.22 | 82.11±1.48 | 70.77±1.35 | 89.51±0.03 | 80.64±0.35 | 65.34±1.70 | 56.37±0.76 | 92.25±0.08 | 54.47±3.42 | 92.60±0.03 | 71.47±2.75 | 0.14±0.10 | 0.22±0.12 |
| GroupDRO | 83.79±0.68 | 58.98±1.84 | 82.60±1.25 | 70.64±1.72 | 89.59±0.09 | 79.21±1.02 | 65.10±1.04 | 56.84±0.63 | 92.29±0.09 | 51.95±2.80 | 92.61±0.03 | 68.24±1.94 | 0.02±0.02 | 0.01±0.04 |
| DANN | 83.90±0.68 | 62.38±2.65 | 81.18±1.37 | 70.63±1.82 | 89.60±0.19 | 79.71±1.35 | 65.22±1.48 | 55.71±1.23 | 92.23±0.08 | 51.46±3.41 | 92.60±0.03 | 65.47±5.35 | 0.12±0.09 | 0.15±0.07 |
| Deep Coral | 84.70±1.17 | 60.11±3.53 | 82.53±1.01 | 68.61±1.70 | 89.68±0.06 | 79.81±0.22 | 64.98±1.03 | 56.14±1.76 | 92.22±0.13 | 53.71±2.75 | 92.61±0.03 | 68.88±3.61 | 0.11±0.15 | 0.09±0.02 |
| DIR | 80.46±0.55 | 57.67±1.41 | 82.54±0.17 | 67.47±2.61 | 84.30±0.46 | 77.65±1.93 | 64.98±1.06 | 56.81±0.91 | 84.53±1.99 | 50.41±5.66 | 87.73±2.60 | 61.50±15.69 | 0.42±0.03 | 0.15±0.03 |
| GIL | 80.02±0.78 | 62.05±1.55 | 82.12±0.52 | 66.18±2.87 | 86.77±1.06 | 78.81±1.35 | 62.05±1.03 | 55.46±1.48 | 83.67±1.54 | 33.20±2.87 | 87.73±2.60 | 38.60±10.58 | 0.14±0.02 | 0.01±0.03 |
| CIGA | 81.65±1.85 | 62.56±1.76 | 81.76±0.35 | 71.47±1.29 | 89.00±0.15 | 81.20±0.75 | 64.10±1.67 | 57.19±1.15 | 90.33±0.97 | 45.36±4.35 | 87.73±2.60 | 45.59±6.44 | 0.30±0.05 | 0.23±0.06 |
| DropNode | 84.09±0.36 | 58.52±0.49 | **83.55±1.07** | 71.18±1.16 | 90.19±0.21 | 81.14±1.73 | 64.56±0.17 | 56.76±0.24 | 91.22±0.11 | 54.14±3.11 | 92.41±0.09 | 74.55±5.56 | -0.02±0.02 | 0.08±0.06 |
| DropEdge | 83.73±0.41 | 59.01±1.90 | 81.63±0.92 | 71.46±1.63 | **90.30±0.18** | 78.93±1.34 | 63.72±0.51 | 57.42±0.48 | 90.07±0.14 | 45.42±1.90 | 82.98±0.99 | 37.69±1.05 | -0.02±0.03 | 0.12±0.05 |
| MaskFeature | 83.44±2.58 | 62.3±3.17 | 83.30±0.45 | 65.90±3.68 | 89.83±0.24 | 82.00±0.73 | 64.92±1.45 | 57.67±1.11 | 92.18±0.01 | 52.24±3.75 | 92.60±0.02 | 64.98±6.95 | -0.02±0.02 | 0.03±0.02 |
| FLAG | **85.37±0.24** | 60.84±2.99 | 82.02±0.67 | 69.11±0.83 | 89.87±0.26 | 77.05±1.27 | 64.74±1.10 | 56.56±0.56 | **92.39±0.04** | 50.85±0.53 | 92.70±0.07 | 66.17±2.87 | 0.02±0.06 | 0.06±0.02 |
| Graph Mixup | 83.16±1.12 | 59.03±3.07 | 82.29±1.34 | 68.88±2.40 | 89.78±0.20 | 80.88±0.60 | 63.54±1.35 | 55.97±1.67 | 92.02±0.10 | 51.48±3.35 | 92.68±0.05 | 70.08±2.06 | -0.08±0.06 | -0.02±0.06 |
| LISA | 83.79±1.04 | 61.50±2.05 | 82.82±1.00 | 71.94±1.31 | 89.36±0.16 | 80.67±0.43 | 63.84±2.15 | 56.14±0.88 | 92.27±0.11 | 53.68±2.65 | **92.71±0.01** | 75.58±0.75 | 0.20±0.02 | 0.05±0.03 |
| G-Mixup | 84.21±1.53 | 61.95±3.15 | 82.83±0.53 | 70.13±2.40 | 89.75±0.17 | 80.28±1.49 | 65.10±1.90 | 56.05±1.76 | 92.19±0.07 | 53.93±3.03 | 92.60±0.00 | 49.27±4.84 | -0.02±0.02 | 0.07±0.04 |
| G-Splice | 84.75±0.18 | 64.46±1.38 | 83.23±0.97 | 72.82±1.16 | 89.71±0.67 | 82.31±0.59 | 64.80±0.92 | 58.02±0.40 | 92.15±1.02 | **86.53±2.66** | 91.92±0.09 | 79.86±13.00 | **0.45±0.09** | 0.37±0.08 |
| G-Splice+R | 84.85±0.19 | **65.56±0.34** | 83.36±0.40 | **73.28±0.16** | 89.10±0.81 | **82.34±0.24** | 64.68±1.15 | **58.34±0.58** | 91.93±0.21 | 85.07±4.50 | 92.14±0.29 | **83.96±7.38** | 0.45±0.04 | **0.40±0.03** |

**Baselines.** We compare our method with both OOD learning algorithms and graph data augmentation methods, as well as the empirical risk minimization (ERM). OOD algorithms include IRM (Arjovsky et al., 2019), VREx (Krueger et al., 2021), DANN (Ganin et al., 2016), Deep Coral (Sun and Saenko, 2016), GroupDRO (Sagawa et al., 2019), and graph OOD methods DIR (Wu et al., 2022b), EERM (Wu et al., 2022a), SRGNN (Zhu et al., 2021), GIL (Li et al., 2022), and CIGA (Chen et al., 2022b). GDA methods include DropNode (Feng et al., 2020), DropEdge (Rong et al., 2019), Feature Masking (Thakoor et al., 2021), FLAG (Kong et al., 2022), Graph Mixup (Wang et al., 2021), LISA (Yao et al., 2022), and G-Mixup (Han et al., 2022). Note that DIR, DropNode, GIL, CIGA, and G-Mixup only apply for graph-level tasks, while EERM and SRGNN only for node-level tasks.

## 6.1 OOD EVALUATION FOR STRUCTURE AND FEATURE EXTRAPOLATION

Structural and feature linear extrapolations can be performed respectively targeting specific types of shifts, as well as combined to solve comprehensive shifts. We demonstrate the superiority of our methods when used concurrently on graph tasks with multiple shifts of structure and feature. Generally, existing OOD datasets construct splits based on selected single shifts. To evaluate on complex graph shifts covering both structure and feature, we create a new dataset, FSMotif. FSMotif is a synthetic dataset where each graph is generated by connecting a base graph and a motif, with the label determined by the motif solely and all nodes assigned color features. We design two complex shift domains, the base graph type combined with color feature, and the graph size combined with color feature. Dataset details are in Appendix H. We report the OOD test accuracy (with OOD validation) of all baselines, as well as G-Splice and FeatX applied both separately and concurrently,

Table 2: OOD performances of all methods on FSMotif. The shifts are size-color and base-color.

| Method | FSMotif-S-C↑ | FSMotif-B-C↑ |
|---|---|---|
| ERM | 35.00(0.98) | 32.67(2.83) |
| IRM | 33.00(0.98) | 36.78(5.67) |
| VREx | 35.33(0.72) | 32.22(2.20) |
| GroupDRO | 32.67(0.82) | 32.22(2.20) |
| DANN | 33.22(0.42) | 31.89(1.73) |
| Coral | 32.66(0.94) | 30.89(0.31) |
| DIR | 33.33(2.16) | 32.11(2.04) |
| GIL | 37.00(0.62) | 23.89(4.50) |
| DropNode | 30.44(1.64) | 43.44(9.75) |
| DropEdge | 32.11(1.40) | 32.22(2.20) |
| FLAG | 34.89(1.23) | 30.67(0.88) |
| GMixup | 31.56(1.25) | 32.45(2.28) |
| Mixup | 31.67(1.41) | 34.78(5.81) |
| Maskfeat | 33.78(0.88) | 37.00(8.96) |
| LISA | 33.89(1.91) | 38.22(10.68) |
| FeatX | 35.89(4.69) | 40.83(6.22) |
| G-Splice | 72.11(6.18) | 50.67(6.40) |
| G-Splice+FeatX | **75.11(2.99)** | **54.11(11.64)** |

across 3 random runs in Table 2. As shown, the combination of G-Splice and FeatX performs evidently better over their separate use and significantly over other baselines, with nearly doubled accuracy on FSMotif-S-C. This strongly evidences that our two strategies can combine to extrapolate over comprehensive shifts, adding to the practicality of our proposed method.

## 6.2 OOD PERFORMANCE ON STRUCTURE SHIFTS

**Datasets & Metrics.** To evidence the generalization improvements of structure extrapolation, we evaluate G-Splice on 8 graph-level OOD datasets with structure shifts. We adopt 5 datasets from the GOOD benchmark (Gui et al., 2022a), HIV-size, HIV-scaffold, SST2-length, Motif-size, and Motif-base, where "-" denotes the shift domain. We construct another natural language dataset Twitter-length (Yuan et al., 2020) following the OOD split of GOOD. Additionally, we adopt protein dataset DD-size and molecular dataset NCI1-size following Bevilacqua et al. (2021). All datasets possess structure shifts, thus proper benchmarks for structural OOD generalization. For evaluation, we report ROC-AUC for GOODHIV, Matthews correlation coefficient (MCC) for DD and NCI1, and accuracy in percentage for all other datasets.

Table 3: Performances of 16 baselines and FeatX on 5 datasets with feature shift. GOODCMNIST is graph-level while others are node-level datasets. "–" denotes the method does not apply on the task.

| feature | GOOD-CMNIST-color↑ | | GOOD-Cora-word↑ | | GOOD-Twitch-language↑ | | GOOD-WebKB-university↑ | | GOOD-CBAS-color↑ | |
|---|---|---|---|---|---|---|---|---|---|---|
| | $ID_{ID}$ | $OOD_{OOD}$ | $ID_{ID}$ | $OOD_{OOD}$ | $ID_{ID}$ | $OOD_{OOD}$ | $ID_{ID}$ | $OOD_{OOD}$ | $ID_{ID}$ | $OOD_{OOD}$ |
| ERM | 77.96±0.34 | 28.60±2.01 | 70.43±0.47 | 64.86±0.38 | 70.66±0.17 | 48.95±3.19 | 38.25±0.68 | 14.29±3.24 | 89.29±3.16 | 76.00±3.00 |
| IRM | 77.92±0.30 | 27.83±1.84 | 70.27±0.33 | 64.77±0.36 | 69.75±0.80 | 47.21±0.98 | 39.34±2.04 | 13.49±0.75 | 91.00±1.28 | 76.00±3.39 |
| VREx | 77.98±0.32 | 28.48±2.08 | 70.47±0.40 | 64.80±0.28 | 70.66±0.18 | 48.99±3.20 | 39.34±1.34 | 14.29±3.24 | 91.14±2.72 | 77.14±1.43 |
| GroupDRO | 77.98±0.38 | 29.07±2.62 | 70.41±0.46 | 64.72±0.34 | 70.84±0.51 | 47.20±0.44 | 39.89±1.57 | 17.20±0.76 | 90.86±2.92 | 76.14±1.78 |
| DANN | 78.00±0.43 | 29.14±2.93 | 70.66±0.36 | 64.77±0.42 | 70.67±0.18 | 48.98±3.22 | 39.89±1.03 | 15.08±0.37 | 90.14±3.16 | 77.57±2.86 |
| Deep Coral | 78.64±0.48 | 29.05±2.19 | 70.47±0.37 | 64.72±0.36 | 70.67±0.28 | 49.64±2.44 | 38.25±1.43 | 13.76±1.30 | 91.14±2.02 | 75.86±3.06 |
| DIR | 31.09±5.92 | 20.60±4.26 | – | – | – | – | – | – | – | – |
| EERM | – | – | 68.79±0.34 | 61.98±0.10 | **73.87**±0.07 | 51.34±1.41 | 46.99±1.69 | 24.61±4.86 | 67.62±4.08 | 52.86±13.75 |
| SRGNN | – | – | 70.27±0.23 | 64.66±0.21 | 70.58±0.53 | 47.30±1.43 | 39.89±1.36 | 13.23±2.93 | 77.62±1.84 | 74.29±4.10 |
| DropNode | **83.51**±0.13 | 33.01±0.12 | – | – | – | – | – | – | – | – |
| DropEdge | 79.51±0.22 | 26.83±0.81 | 71.03±0.15 | 65.55±0.29 | 70.66±0.10 | 47.94±3.42 | 38.79±0.77 | 14.28±2.34 | 84.29±1.16 | 77.62±3.37 |
| MaskFeature | 78.32±0.63 | 44.85±2.42 | 70.99±0.22 | 64.42±0.35 | 70.58±0.80 | 48.61±3.95 | 39.89±0.77 | 15.08±1.30 | 90.48±7.50 | 77.62±1.35 |
| FLAG | 79.05±0.41 | 37.74±7.88 | 70.59±0.11 | 64.95±0.41 | 70.54±0.62 | 45.66±1.17 | 37.70±4.01 | 12.70±2.33 | 91.90±2.69 | 81.43±1.17 |
| Graph Mixup | 77.40±0.22 | 26.47±1.73 | **71.54**±0.63 | 65.23±0.56 | 71.30±0.14 | 52.27±0.78 | **54.65**±3.41 | 17.46±1.94 | 73.57±8.72 | 70.57±7.41 |
| LISA | 76.75±0.46 | 29.63±2.82 | 70.15±0.20 | 64.96±0.17 | 71.09±0.53 | 45.55±0.55 | 37.70±0.00 | 16.40±2.62 | **94.29**±1.16 | 83.34±1.35 |
| G-Mixup | 77.58±0.29 | 26.40±1.47 | – | – | – | – | – | – | – | – |
| FeatX | 69.54±1.51 | **62.49**±2.12 | 70.39±0.36 | **66.12**±0.54 | 70.94±0.37 | **52.76**±0.23 | 50.82±0.00 | **32.54**±8.98 | 92.86±1.17 | **87.62**±2.43 |

**Results.** OOD performances of G-Splice and all baselines on structure distribution shifts are shown in Table 1. As can be observed, G-Splice consistently outperforms all other methods in OOD test results, showing effectiveness in various structural OOD tasks. On synthetic dataset GOODMotif, G-Splice substantially outperforms most baselines by 60% for size domain and 20% for base domain in accuracy, approaching ID performances, which evidences the generalization improvements achieved by our structural extrapolation. With a VREx-like regularization applied, G-Splice+R achieve further performance gain on most datasets, implying that combined use of augmented environment information with both data extrapolation and invariance regularization is beneficial. Furthermore, in contrast to OOD performances, G-Splice does not always perform best in ID settings. Also, G-Splice shows significant performance gain compared with other graph data augmentation methods. This reveals that G-Splice enhances generalization abilities with extrapolation strategies rather than overall progress in learning or simple data augmentation. By guiding the model to extrapolate with OOD samples, G-Splice extends the data distribution and improves generalization for specific structure shifts. As ablation studies, we evidence that certain extrapolation procedures specifically benefit size or base shifts, supporting our theoretical analysis, which is detailed in Appendix I.

### 6.3 OOD Performance on Feature Shifts

**Datasets & Metrics** To show the OOD effectiveness of feature extrapolation, we evaluate FeatX on 5 graph OOD datasets with feature shifts. We adopt 5 datasets from the GOOD benchmark, CMNIST-color, Cora-word, Twitch-language, WebKB-university, and CBAS-color, with more details in Appendix H. All shift domains are structure-irrelevant and provide specific evaluation on features. We report accuracy in percentage for all 5 datasets.

**Results.** OOD performances of FeatX and all baselines on feature shifts are shown in Table 3. We can observe that FeatX consistently outperforms all other methods in OOD test results, showing its effectiveness in various feature OOD tasks. On GOODWebKB, FeatX substantially outperforms most baselines by 100% in accuracy. On synthetic dataset GOODCBAS, FeatX outperforms most baselines by 14% and achieves OOD results close to ID results, substantially solving the feature shift with feature extrapolation. FeatX does not always outperform in ID settings; also, FeatX shows significant performance gain compared with other graph data augmentation methods. This reveals that FeatX specifically enhances generalization abilities in feature rather than making overall progress in learning with simple data augmentation. FeatX succeeds in selecting non-causal features and lead the model to extrapolate with OOD samples spanning the selected feature space.

## 7 Discussion

Our work introduces an innovative GDA approach to solve graph OOD generalization using linear extrapolation in graph space. Our environment-aware framework, featuring G-Splice and FeatX, improves over existing methods by generating causally-valid OOD samples that enhance model performance. Overall, our data-centric approach opens a new direction in graph OOD studies. Currently, our method depends on the quality of environment information and GNN expressiveness. Also, our theoretical analysis focus on linear extrapolation. Future works could explore optimizing environment information and incorporating non-linear extrapolation studies.

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

# A BACKGROUND AND RELATED WORKS

**Out-of-Distribution (OOD) Generalization.** Out-of-Distribution (OOD) Generalization (Shen et al., 2021; Duchi and Namkoong, 2021; Shen et al., 2020; Liu et al., 2021a) addresses the challenge of adapting a model, trained on one distribution (source), to effectively process unseen data from a potentially different distribution (target). It shares strong ties with various areas such as transfer learning (Weiss et al., 2016; Torrey and Shavlik, 2010; Zhuang et al., 2020), domain adaptation (Wang and Deng, 2018), domain generalization (Wang et al., 2022), causality (Pearl, 2009; Peters et al., 2017), and invariant learning (Arjovsky et al., 2019). As a form of transfer learning, OOD generalization is especially challenging when the target distribution substantially differs from the source distribution. OOD generalization, also known as distribution or dataset shift (Quiñonero-Candela et al., 2008; Moreno-Torres et al., 2012), encapsulates several concepts including covariate shift (Shimodaira, 2000), concept shift (Widmer and Kubat, 1996), and prior shift (Quiñonero-Candela et al., 2008). Both Domain Adaptation (DA) and Domain Generalization (DG) can be viewed as specific instances of OOD, each with its own unique assumptions and challenges.

**Domain Generalization (DG).** DG (Wang et al., 2022; Li et al., 2017; Muandet et al., 2013; Deshmukh et al., 2019; Gui et al., 2020) strives to predict samples from unseen domains without the need for pre-collected target samples, making it more practical than DA in many circumstances. However, generalizing without additional information is logically implausible, a conclusion also supported by the principles of causality (Pearl, 2009; Peters et al., 2017). As a result, contemporary DG methods have proposed the use of domain partitions (Ganin et al., 2016; Zhang et al., 2022) to generate models that are domain-invariant. Yet, due to the ambiguous definition of domain partitions, many DG methods lack robust theoretical underpinning.

**Causality & Invariant Learning.** Causality (Peters et al., 2016; Pearl, 2009; Peters et al., 2017) and invariant learning (Arjovsky et al., 2019; Rosenfeld et al., 2020; Ahuja et al., 2021) provide a theoretical foundation for the above concepts, offering a framework to model various distribution shift scenarios as structural causal models (SCMs). SCMs, which bear resemblance to Bayesian networks (Heckerman, 1998), are underpinned by the assumption of independent causal mechanisms, a fundamental premise in causality. Intuitively, this supposition holds that causal correlations in SCMs are stable, independent mechanisms akin to unchanging physical laws, rendering these causal mechanisms generalizable. An assumption of a data-generating SCM equates to the presumption that data samples are generated through these universal mechanisms. Hence, constructing a model with generalization ability requires the model to approximate these invariant causal mechanisms. Given such a model, its performance is ensured when data obeys the underlying data generation assumption. Peters et al. (2016) initially proposed optimal predictors invariant across all environments (or interventions). Motivated by this work, Arjovsky et al. (2019) proposed framing this invariant prediction concept as an optimization process, considering one of the most popular data generation assumptions, PIIF. Consequently, numerous subsequent works (Rosenfeld et al., 2020; Ahuja et al., 2021; Chen et al., 2022a; Lu et al., 2021)—referred to as invariant learning—considered the initial intervention-based environments (Peters et al., 2016) as an environment variable in SCMs. When these environment variables are viewed as domain indicators, it becomes evident that this SCM also provides theoretical support for DG, thereby aligning many invariant works with the DG setting. Besides PIIF, many works have considered FIIF and anti-causal assumptions (Rosenfeld et al., 2020; Ahuja et al., 2021; Chen et al., 2022a), which makes these assumptions as popular basics of causal theoretical analyses.

**OOD generalization for graph.** Extrapolating on non-Euclidean data has garnered increased attention, leading to a variety of applications (Sanchez-Gonzalez et al., 2018; Barrett et al., 2018; Saxton et al., 2019; Battaglia et al., 2016; Tang et al., 2020; Veličković et al., 2019; Xu et al., 2019b). Inspired by Xu et al. (2020), Yang et al. (2022a) proposed that GNNs intrinsically possess superior generalization capability. Several prior works (Knyazev et al., 2019; Yehudai et al., 2021; Bevilacqua et al., 2021) explored graph generalization in terms of graph sizes, with Bevilacqua et al. (2021) being the first to study this issue using causal models. Recently, causality modeling-based methods have been proposed for both graph-level tasks (Wu et al., 2022b; Miao et al., 2022; Chen et al., 2022b; Fan et al., 2022; Yang et al., 2022b) and node-level tasks (Wu et al., 2022a). To solve OOD problems in graph, DIR (Wu et al., 2022b) selects graph representations as causal rationales and conducts causal intervention to create multiple distributions. EERM (Wu et al., 2022a) generates environments with REINFORCE algorithm to maximize loss variance between

environments while adversarially minimizing the loss. SRGNN (Zhu et al., 2021) aims at pushing biased training data to the given unbiased distribution, performed through central moment discrepancy and kernel matching. To improve interpretation and prediction, GSAT (Miao et al., 2022) learns task-relevant subgraphs by constraining information with stochasticity in attention weights. CIGA (Chen et al., 2022b) models the graph generation process and learns subgraphs to maximally preserve invariant intra-class information. GREA (Liu et al., 2022) performs rationale identification and environment replacement to augment virtual data examples. GIL (Li et al., 2022) proposes to identify invariant subgraphs and infer latent environment labels for variant subgraphs through joint learning. However, except for CIGA (Chen et al., 2022b), their data assumptions are less comprehensive compared to traditional OOD generalization. CIGA, while recognizing the importance of diverse data generation assumptions (SCMs), attempts to fill the gap through non-trivial extra assumptions without environment information. Additionally, environment inference methods have gained traction in graph tasks, including EERM (Wu et al., 2022a), MRL (Yang et al., 2022b), and GIL (Li et al., 2022). However, these methods face two undeniable challenges. First, their environment inference results require environment exploit methods for evaluation, but there are no such methods that perform adequately on graph tasks according to the synthetic dataset results in GOOD benchmark (Gui et al., 2022a). Second, environment inference is essentially a process of injecting human assumptions to generate environment partitions, but these assumptions are not well compared.

**Graph data augmentation for generalization.** Some data augmentation methods, not limited to graph methods, empirically show improvements in OOD generalization tasks. Mixup (Zhang et al., 2017), which augments samples by interpolating two labeled training samples, is reported to benefit generalization. LISA (Yao et al., 2022) selectively interpolates intra-label or intra-domain samples to further improve OOD robustness. In the graph area, following Mixup, Graph Mixup (Wang et al., 2021) mixes the hidden representations in each GNN layer, while ifMixup (Guo and Mao, 2021) directly applies Mixup on the graph data instead of the latent space. Graph Transplant (Park et al., 2022) employs node saliency information to select a substructure from each graph as units to mix. G-Mixup (Han et al., 2022) interpolates the graph generator of each class and mixes on class-level to improve GNN robustness. DPS (Yu et al., 2022) extracts multiple label-invariant subgraphs with a set of subgraph generators to train an invariant GNN predictor. However, few works target OOD problems, and no prior work generates OOD samples that can provably generalize over graph distribution shifts. In contrast, we offer a graph augmentation method to extrapolate in structure and feature for OOD generalization.

**Technical comparisons with prior methods.** We discuss in detail the technical differences between existing works and ours. DIR and GREA algorithms are much alike by design, identifying causal subgraphs and switching non-causal subgraphs between graphs. With this localized strategy, their augmented environments can only cover local base shifts, leaving the global structural extrapolation unexplored. EERM exclusively considers node-level tasks, and only performs edge addition/deletion to cover minor shifts on graph base. GDA methods GMixup and Graph Transplant provide no guarantee for solving OOD related tasks, and can not deal with global structure shifts such as size. LiSA (Yu et al., 2023) extracts multiple subgraphs and AdvCA (Sui et al., 2022) masks certain nodes/edges from given graphs to generate graph augmentations. SizeShiftReg (Buffelli et al., 2022) uses coarsening to extracts multiple subgraphs from given graphs, obtaining slightly smaller augmented graphs (80% or 90% of the original graph in actual implementation). These strategies result in augmented graphs that only contain smaller substructures, restricting their potential extrapolation to one instead of both distribution directions. In this case, a common test scenario where test graphs are larger than the training graphs is not covered. Mixup and ExtraMix (Kwon et al., 2022) apply strategies on feature levels without designs for graph structure. In contrast, we study non-Euclidean space extrapolation in a far more systematic way. Our method considers the completeness of achieving both feature and structural extrapolation, and further cover structural global/local extrapolation (or size/base shifts by example) in both distribution directions. This substantially sets the difference between our method and the existing works. Moreover, our novel theoretical contributions include proposing non-Euclidean space linear extrapolation with definitions, analyses, and guarantees. Considering techniques, our design of graph splice serves global extrapolation and avoids add-on nodes to preserve graph structures, divergent from linker design approaches for molecules (Huang et al., 2022; Igashov et al., 2022). In addition, we design subgraph extraction by label-environment-aware pair learning, a novel technique over previous studies.

## B   COMPUTATIONAL COMPLEXITY ANALYSIS

We provide the theoretical analysis regarding the time and space computational complexity as follows. For computation, we generally use one NVIDIA GeForce RTX 2080 Ti for each single experiment.

The time complexity of our G-Splice is $O((|V|^2 d + |V|d^2)|B|)$, where $|V|$ denotes the number of nodes, $|B|$ is the batch size, and $d$ is the dimensionality of the representations. The time complexity of our FeatX is $O((|E|d + |V|d^2)|B|)$, where $|E|$ denotes the number of edges. Specifically, message-passing GNNs has a complexity of $O((|E|d + |V|d^2)|B|)$, which we adopt to instantiate our GNN components. For G-Splice, the time complexity of obtaining GNN representations is $O((|E|d + |V|d^2)|B|)$, and that of pair-wise similarity matching and bridge generation are both $O(|V|^2 d|B|)$. Since $O(|E|) \leq O(|V|^2)$, the overall time complexity of G-Splice is $O((|E|d + |V|^2 d + |V|d^2)|B|) = O(((|V|^2 d + |V|d^2)|B|)$. For FeatX, the time complexity of non-causal feature selection and feature extrapolation are both $O(|V|d_V|B|)$, where $d_V$ is the node feature dimension, which is far smaller than the time complexity of GNN representations. In comparison, the time complexity of most GNN-based graph representation methods are $O(|E|d + |V|d^2)$, including simple algorithms like ERM, IRM, and VRex implemented with GNNs. More complicated graph methods such as CIGA has time complexity $O(((|V|^2 d + |V|d^2)|B|)$, which is also the case for G-Splice. Therefore, the time complexity of our proposed methods is on par with the existing methods.

The space complexity of G-Splice and FeatX is $O(|V|dL + |E|d)$, where $L$ is the number of GNN layers. This is the space complexity of the message-passing GNNs we use, as well as the space complexity of most GNN-based methods.

## C   TECHNICAL DETAILS

We complete the technical details of causal and environmental subgraph extractions here. Let $G_{\varepsilon_1}$ and $G_{\varepsilon_2}$ be two graphs with the same label $y_1$ but different environments $\varepsilon_1$ and $\varepsilon_2$. As we have discussed, $G_{\text{inv}}$ should be the subgraph both graphs contain and have most in common. Since graph neural networks aggregate information of an ego graph, i.e., the local subgraph within $k$-hop of a node, to the embedding of that node through message passing, nodes with similar ego graphs should have similar embeddings. Therefore, in the node embedding space, nodes from $G_{\varepsilon_1}$ and $G_{\varepsilon_2}$ with similar representations should be a part of $G_{\text{inv}}$. We encode both graphs into node embeddings with a GNN and calculate their weighted similarity matrix $\boldsymbol{S^w}$, each element of which is the weighted cosine similarity of a pair of nodes from $G_{\varepsilon_1}$ and $G_{\varepsilon_2}$, i.e.,

$$\boldsymbol{S^w_{ij}} = w * S_c(\boldsymbol{z}_i, \boldsymbol{z}_j), \text{ for } v_i \sim G_{\varepsilon_1}, v_j \sim G_{\varepsilon_2}, \tag{5}$$

where $w$ is a trainable parameter and $S_c(\cdot)$ is the cosine similarity calculation. The scores in the weighted similarity matrix $\boldsymbol{S^w}$ are considered as probabilities to sample the causal subgraph from either $G_{\varepsilon_1}$ or $G_{\varepsilon_2}$. We use label-invariant and environment-variant graph pairs to pre-train a causal subgraph searching network, which is optimized for the sampled causal subgraphs to be capable of predicting the label $Y$ solely.

Similarly, we perform similarity matching for environment-invariant graph pairs to extract environmental subgraphs $G_{\text{env}}$ that are determined by the environment $\mathcal{E}$. Graphs from the same environment should contain similar subgraphs, and we aim at extracting these environmental subgraphs. Environment-invariant and label-variant graph pairs are used to pre-train the environmental subgraph searching network. We calculate the weighted similarity scores from embeddings and sample subgraphs with probabilities. The network is optimized using the environment label $\varepsilon$ for the sampled subgraphs to predict the environment.

## D   FURTHER DISCUSSIONS

### D.1   CONNECTIONS BETWEEN SLE AND FLE

The connection between feature extrapolation and structural extrapolation is implicitly described in Sec 2's Graph structure and feature distribution shifts and Sec 3 Linear Extrapolation in Graph Space. Firstly, Linear Extrapolation, which constructs samples beyond the known range while maintaining the same direction and magnitude of known sample differences, is a central concept in

our approach. Graph data are complex in that it contains features as well as topological structures. We propose to define Linear Extrapolation for both feature and structure (Sec 3.2), and together they form the complete definition of Graph Linear Extrapolation. Though their definition have different variables, the formulas share the common form of mathematical organization as linear extrapolations. Secondly, graph distribution shifts can happen on both features and structures, which possesses different properties and should be handled separately. While feature extrapolation and structural extrapolation can be used to solve respective shifts, when combined they can address complex feature-structure shifts. Therefore, they complement each other in a systematic solution of graph OOD problems. Thirdly, feature and structural extrapolation share similar logic in causal analyses and theoretical justification. Combined causal analyses are given in Sec 3.1. In Sec 3.3, we provide theoretical guarantees that structural linear extrapolation has the capability to generate OOD samples that are both plausible and diverse, while the justification of feature linear extrapolation is relatively straightforward and provided in Section 5.3.

## D.2    APPLICABILITY OF THE CAUSAL ADDITIVITY ASSUMPTION

Our work builds upon the principle of Causal Additivity, a causal assumption widely applicable in graph classification tasks. This assumption can be subjectively verified through the logic for common natural language sentimental analysis datasets such as SST2 and Twitter, as well as synthetic dataset GOOD-Motif, where labels are determined by certain structures. The spliced graph contains combined causal structures; therefore, it forms a causally valid sample when given the mixed label of all component graphs. For molecule/protein datasets with chemical property tasks, the assumption is strongly underpinned by experimental results, as evidenced by the improved or comparable results even when whole samples are randomly combined in Appendix E.1.1. Although we acknowledge that our assumption may not encompass all cases, it does make headway in addressing a substantial class of problems. As graph OOD generalization is a complex issue in practice, different techniques are required for varying domains and problems. No single method can be expected to resolve all unknown cases, and our future work aims to expand the scope of tasks addressed.

## D.3    STRENGTHS OF LINEAR EXTRAPOLATION OVER INTERPOLATION

Data augmentation methods can introduce additional samples not covered by the training database to benefit model learning. Since we focus on tasks that are out-of-distribution instead of in-distribution, models are expected to extrapolate instead of interpolate to make predictions outside the training range. However, the distribution area where models cannot generalize to is also hardly reachable when generating augmentation samples using traditional interpolation techniques. Interpolation methods cannot provide any guarantees regarding solving graph distribution shifts. In contrast, theoretical and empirical analyses show that linear extrapolation can generalize over certain shifts. Specifically, it can be reasoned based on our theoretical studies. Theorem 5.1 establishes that feature linear extrapolation can achieve invariant prediction and generalize over distribution shifts regarding selected variant features under certain environment conditions. Theorem 3.1 establishes that structural linear extrapolation can create OOD samples covering at least two environments in opposite directions of the distribution, respecting size and base shifts each. Contrarily, in the case of interpolation, the constructions in the proofs would not hold, thus failing to build these guarantees.

## D.4    METHOD APPLICABILITY WHEN THE DISTRIBUTION SHIFT TYPE IS UNKNOWN

Firstly, as we have evidenced, our two strategies for feature and structure can be combined to solve comprehensive shifts. Also, our proposed method is applicable when OOD knowledge of a dataset is not fully known, since we are able to choose the techniques as hyperparameter selection. This allows the framework to automatically decide on using either method separately or combined, thus covering OOD tasks with structural, feature or complex shifts. Secondly, although we use specific expressions of base and size shifts in theoretical studies of structural shifts, discussions for base and size extrapolation are actually applicable to general local and global structural extrapolation, respectively, in the field of graph. Therefore, G-Splice can cover both local and global structural shifts, making it applicable towards various unknown distribution shifts. One potential evidence is that it performs favorably on multiple real-world datasets, which inevitably contain natural and unknown distribution shifts.

## E   BROADER IMPACTS

Addressing out-of-distribution (OOD) generalization presents a formidable challenge, particularly in the realm of graph learning. This issue is acutely exacerbated when conducting scientific experiments becomes cost-prohibitive or impractical. In many real-world scenarios, data collection is confined to certain domains, yet extrapolating this knowledge to broader areas, where experiment conduction proves difficult, is crucial. In focusing on a data-centric approach to the OOD generalization problem, we pave the way for the integration of graph data augmentation with graph OOD, a strategy with substantial potential for broad societal and scientific impact.

Our research adheres strictly to ethical guidelines and does not raise any ethical issues. It neither involves human subjects nor gives rise to potential negative social impacts or privacy and fairness issues. Furthermore, we foresee no potential for malicious or unintended usage of our work. Nonetheless, we acknowledge that all technological progress inherently carries risks. Consequently, we advocate for ongoing evaluation of the broader implications of our methodology across a range of contexts.

## F   LIMITATIONS

Our work builds upon the principle of Causal Additivity, a causal assumption widely applicable in graph classification tasks. This assumption can be verified theoretically and experimentally for a variety of graph classification tasks. However, we acknowledge that our assumption may not encompass all cases. As graph OOD generalization is a complex issue in practice, different domains and problems may required varying techniques and our method might not resolve all unknown cases. Our future work aims to expand the scope of tasks addressed.

For another, our work discusses shifts on both graph structure and feature. By respective considerations, while G-splice can solve structure shifts, it augments structural OOD samples, which creates additional shift when facing feature-OOD situations. FeatX stands in the similar situation, introducing extra shifts for structural OOD problems. When combined, G-Splice and FeatX can solve both types of shifts and is suitable for addressing complex OOD situations. As "all medicine has its side effects", their concurrent use would create extra shifts if the problem does not involve both shifts. Given that an OOD dataset only contain one type between structure/feature shifts, the performance gain might not be so ideal when using two methods concurrently. However, this does not impair their applicability when OOD knowledge of the dataset is not fully known, since we are able to choose the techniques similarly as hyperparameter selection.

In addition, our current work does not discuss link prediction, which is an important task in graph learning. Thus our future work aims to expand the scope of tasks addressed. Furthermore, the proposed methods are designed to cover complex shifts of multiple types, therefore the hyperparameter selections including selection of techniques require certain amounts of pre-computation, which sets prerequisites in computational resources.

## G   THEORETICAL PROOFS

This section presents comprehensive proofs for all the theorems mentioned in this paper, along with the derivation of key intermediate results and necessary discussions.

**Theorem 3.1** *Given an $N$-sample training dataset $D\{G_{tr}\}$, its $N$-dimension structural linear extrapolation can generate sets $D\{G_1\}$ and $D\{G_2\}$ s.t. $(G_1)_{env} < (G_{tr})_{env} < (G_2)_{env}$ for $\forall G_{tr}, G_1, G_2$, where $<$ denotes "less in size" for size extrapolation and "lower base complexity" for base extrapolation.*

*Proof.* Considering size extrapolation, we prove that 1.sets $D\{G_1\}$ and $D\{G_2\}$ contain graph sizes achievable by N-dimension structural linear extrapolation; 2.$|\boldsymbol{X}|_{G_1} < |\boldsymbol{X}|_{G_{tr}} < |\boldsymbol{X}|_{G_2}$ holds for $\forall G_{tr}, G_1, G_2$.

For N-dimension structural linear extrapolation on training data $D\{G_{tr}\}$, we have Eq. 2:

$$G_{sle}^N = \sum_{i=1}^{N} a_i \cdot G_i + \sum_{i=1}^{N} \sum_{j=1}^{N} b_{ij} \cdot (G_j - G_i) = \mathbf{a}^\top \mathbf{G} + \langle B, \mathbf{1}\mathbf{G}^\top - \mathbf{G}\mathbf{1}^\top \rangle_F.$$

Let the largest and smallest graph $G_{ma}$ and $G_{mi}$ in $D\{G_{tr}\}$ be indexed $i = ma$ and $i = mi$. We generate $D\{G_2\}$ using Eq. 2 with the condition that $a_{ma} = 1$ and $\sum_{i=1}^{N} a_i \geq 2$. We generate $D\{G_1\}$ with the condition that $\sum_{i=1}^{N} a_i = 0$, $\sum_{i=1}^{N} \sum_{j=1}^{N} b_{ij} = 1$ and $b_{(mi)j} = 1$. By Definition 3, $D\{G_1\}$ and $D\{G_2\}$ contain graph sizes achievable by N-dimension structural linear extrapolation.

For $\forall G_2 \in D\{G_2\}$, since $a_{ma} = 1$ and $\sum_{i=1}^{N} a_i \geq 2$, $G_2$ contains multiple graphs spliced together; then we have

$$|\boldsymbol{X}|_{G_2} > |\boldsymbol{X}|_{G_{ma}} \geq |\boldsymbol{X}|_{G_{tr}} \tag{6}$$

for $\forall G_{tr} \in D\{G_{tr}\}$. For $\forall G_1 \in D\{G_1\}$, since $\sum_{i=1}^{N} a_i = 0$, $\sum_{i=1}^{N} \sum_{j=1}^{N} b_{ij} = 1$ and $b_{(mi)j} = 1$, $G_1$ contains only one single subgraph extracted from $G_{mi}$ and another graph; then we have

$$|\boldsymbol{X}|_{G_1} < |\boldsymbol{X}|_{G_{mi}} \leq |\boldsymbol{X}|_{G_{tr}} \tag{7}$$

for $\forall G_{tr} \in D\{G_{tr}\}$. Therefore, $|\boldsymbol{X}|_{G_1} < |\boldsymbol{X}|_{G_{tr}} < |\boldsymbol{X}|_{G_2}$ holds for $\forall G_{tr}, G_1, G_2$.

Considering base extrapolation, we prove that 1.sets $D\{G_1\}$ and $D\{G_2\}$ contain graph bases achievable by N-dimension structural linear extrapolation; 2.$\mathcal{B}_{G_1} < \mathcal{B}_{G_{tr}} < \mathcal{B}_{G_2}$ holds for $\forall G_{tr}, G_1, G_2$, where $\mathcal{B}$ denotes the base graph and "$<$" denotes less complex in graph base. Note that graph bases can be numerically indexed for ordering and comparisons, such as the Bemis-Murcko scaffold algorithm (Bemis and Murcko, 1996).

For N-dimension structural linear extrapolation on training data $D\{G_{tr}\}$, following Eq. 2, let the graphs with the most and least complex graph base $G_{mo}$ and $G_{le}$ in $D\{G_{tr}\}$ be indexed $i = mo$ and $i = le$. We generate $D\{G_2\}$ using Eq. 2 with the condition that $a_{mo} = 1$ and $\sum_{i=1}^{N} a_i \geq 2$. We generate $D\{G_1\}$ with the condition that $\sum_{i=1}^{N} a_i = 0$, $\sum_{i=1}^{N} \sum_{j=1}^{N} b_{ij} = 1$ and $b_{(le)j} = 1$, with $(G_j - G_i)$ being a causal graph extraction. By Definition 4, $D\{G_1\}$ and $D\{G_2\}$ contain graph bases achievable by N-dimension structural linear extrapolation.

For $\forall G_2 \in D\{G_2\}$, since $a_{mo} = 1$ and $\sum_{i=1}^{N} a_i \geq 2$, $G_2$ contains multiple graphs spliced together including the most complex base; then we have

$$\mathcal{B}_{G_2} > \mathcal{B}_{G_{mo}} \geq \mathcal{B}_{G_{tr}} \tag{8}$$

for $\forall G_{tr} \in D\{G_{tr}\}$, adding upon $\mathcal{B}_{G_{mo}}$ to create more complex base graphs. For $\forall G_1 \in D\{G_1\}$, since $\sum_{i=1}^{N} a_i = 0$, $\sum_{i=1}^{N} \sum_{j=1}^{N} b_{ij} = 1$ and $b_{(le)j} = 1$ with $(G_j - G_i)$ being a causal graph extraction, $G_1$ contains only a single causal subgraph extracted from $G_{le}$; then we have

$$\mathcal{B}_{G_1} < \mathcal{B}_{G_{le}} \leq \mathcal{B}_{G_{tr}} \tag{9}$$

for $\forall G_{tr} \in D\{G_{tr}\}$, essentially creating structural linear extrapolations containing no base graphs. Therefore, $\mathcal{B}_{G_1} < \mathcal{B}_{G_{tr}} < \mathcal{B}_{G_2}$ holds for $\forall G_{tr}, G_1, G_2$.

This completes the proof. □

**Theorem 3.2** *Given an $N$-sample training dataset $D\{G_{tr}\}$ and its true labeling function for the target classification task $f(G)$, if $D\{G_{sle}^N\}$ is a graph set sampled from the N-dimension structural linear extrapolation of $D\{G_{tr}\}$ and Assumption 1 holds, then for $\forall (G_{sle}^N, y) \in D\{G_{sle}^N\}$, $y = f(G_{sle}^N)$.*

*Proof.* By Definition 2, for N-dimension structural linear extrapolation on training data $D\{G_{tr}\}$, for $\forall (G_{sle}^N, y)$ we have $G_{sle}^N$

$$G_{sle}^N = \sum_{i=1}^{N} a_i \cdot G_i + \sum_{i=1}^{N} \sum_{j=1}^{N} b_{ij} \cdot (G_j - G_i) = \mathbf{a}^\top \mathbf{G} + \langle B, \mathbf{1}\mathbf{G}^\top - \mathbf{G}\mathbf{1}^\top \rangle_F,$$

and the label $y$ for $G_{sle}^N$

$$y = (\sum_{i=1}^{N} a_i \cdot y_i + \sum_{i=1}^{N} \sum_{j=1}^{N} c_{ij} b_{ij} \cdot y_j) / (\sum_{i=1}^{N} a_i + \sum_{i=1}^{N} \sum_{j=1}^{N} c_{ij} b_{ij})$$
$$= (\mathbf{a}^\top \mathbf{y} + \langle C \circ B, \mathbf{1}\mathbf{y}^\top \rangle_F) / (\mathbf{a}^\top \mathbf{1} + \langle C, B \rangle_F).$$

$\mathbf{a}^\top \mathbf{G}$ splices $\sum_{i=1}^{N} a_i$ graphs together, and since $[G_1, \ldots, G_N]$ are the $N$ graphs from $D\{G_{tr}\}$, each of $G_i$ contains one and only one causal graph. Under the causal additivity of Assumption 1, given $G' = G_1 + G_2$, we have $f(G') = ay_1 + (1-a)y_2$. With a fair approximation of $a = 1 - a = 1/2$, we can feasibly obtain $f(G') = (y_1 + y_2)/2$. Recursively, for $\mathbf{a}^\top \mathbf{G}$ we can derive

$$f(\mathbf{a}^\top \mathbf{G}) = (\sum_{i=1}^{N} a_i \cdot y_i) / (\sum_{i=1}^{N} a_i). \tag{10}$$

$\langle B, \mathbf{1}\mathbf{G}^\top - \mathbf{G}\mathbf{1}^\top \rangle_F$ splices $\sum_{i=1}^{N} \sum_{j=1}^{N} b_{ij}$ extracted subgraphs together. Among them, $\sum_{i=1}^{N} \sum_{j=1}^{N} c_{ij} b_{ij}$ are causal subgraphs, while the others are environmental subgraphs. Similarly, using the causal additivity of Assumption 1 in a recursive manner, we can derive

$$f(\langle B, \mathbf{1}\mathbf{G}^\top - \mathbf{G}\mathbf{1}^\top \rangle_F) = (\sum_{i=1}^{N} \sum_{j=1}^{N} c_{ij} b_{ij} \cdot y_i) / (\sum_{i=1}^{N} \sum_{j=1}^{N} c_{ij} b_{ij}). \tag{11}$$

Combining the results of Eq. 10 and Eq. 11, using Assumption 1 in a recursive manner, we can derive for $\mathbf{a}^\top \mathbf{G} + \langle B, \mathbf{1}\mathbf{G}^\top - \mathbf{G}\mathbf{1}^\top \rangle_F$:

$$f(\mathbf{a}^\top \mathbf{G} + \langle B, \mathbf{1}\mathbf{G}^\top - \mathbf{G}\mathbf{1}^\top \rangle_F) = (\sum_{i=1}^{N} a_i \cdot y_i + \sum_{i=1}^{N} \sum_{j=1}^{N} c_{ij} b_{ij} \cdot y_j) / (\sum_{i=1}^{N} a_i + \sum_{i=1}^{N} \sum_{j=1}^{N} c_{ij} b_{ij}). \tag{12}$$

By Definition 2, we have

$$f(G_{sle}^N) = f(\mathbf{a}^\top \mathbf{G} + \langle B, \mathbf{1}\mathbf{G}^\top - \mathbf{G}\mathbf{1}^\top \rangle_F)$$
$$= (\sum_{i=1}^{N} a_i \cdot y_i + \sum_{i=1}^{N} \sum_{j=1}^{N} c_{ij} b_{ij} \cdot y_j) / (\sum_{i=1}^{N} a_i + \sum_{i=1}^{N} \sum_{j=1}^{N} c_{ij} b_{ij}) = y.$$

Therefore, for $\forall (G_{sle}^N, y) \in D\{G_{sle}^N\}$, we have $y = f(G_{sle}^N)$.

This completes the proof. $\square$

**Theorem 4.1** *Given the causal graph (Figure 1(b)) and assuming a bijective causal mapping between $C$ and $Y$, for two same-class different-environment graphs $G = G_{y,\epsilon}$ and $G' = G_{y,\epsilon'}$, let $G_s \subseteq G$ and $G'_s \subseteq G'$ represent the subgraphs of $G$ and $G'$. Let $I(\cdot, \cdot) \in [0, 1]$ a similarity function in the graph hidden feature space and $f_z : \mathcal{G} \to \mathbb{R}^f$ is a feature mapping that reversely infers hidden features, e.g., $C$ and $S_1$, then:*

*(1) Given the invariant subgraphs of $G$ and $G'$, $G_{inv}$ and $G'_{inv}$, the similarity of the subgraphs defined in the corresponding graph feature space can be represented as $I(f_z(G_{inv}), f_z(G'_{inv}))$. It follows that the value of this similarity reaches its maximum 1;*

*(2) Given a subgraph set $\mathbf{G}_s = \{G_s | \exists G'_s, I(f_z(G_s), f_z(G'_s)) = 1\}$, the invariant subgraph $G_{inv}$ of $G$ can be obtained by optimizing the objective: $G_{inv} = \mathrm{argmax}_{G_s \in \mathbf{G}_s} |G_s|$.*

*Proof.* The invariant subgraphs $G_{inv}$ in the same class $Y$ share the same features inferred by $f_z$ because of the bijective causal mapping assumption. This implies that the similarity of invariant subgraphs in the same class is able to reach the maximal value of 1. In contrast, subgraphs affected by $S_1$ and $S_2$ do not have this property since $S_1$ and $S_2$ are noises fluctuating frequently, leading to diverse $G_{s1}$ and $G_{s2}$ that are assumed not to be matched in different graphs. Concretely, this mismatching is caused by the differences in feature dimensions corresponding to $S_1$ and $S_2$, which

leads to similarities strictly lower than 1, *i.e.*, $\forall G_s, \exists G_n \subseteq G_s, G_n \neq \emptyset$ s.t. $G_n \in G_{s1} \cup G_{s2} \Leftrightarrow \forall G'_s, I(f_z(G_s), f_z(G'_s)) < 1$. Note that our causal graph is the general case covering the common SCMs of covariate shift, FIIF, and PIIF assumptions when latent variable $S_2$ is not considered.

**Proof of (1):** According to the aforementioned assumption, since $G$ and $G'$ have the same label, it follows that $f_z(G_{\text{inv}}) = f_z(G'_{\text{inv}})$, which directly results in $I(f_z(G_{\text{inv}}), f_z(G'_{\text{inv}})) = 1$

**Proof of (2):** First, given the subgraph set $\mathbf{G}_s$, it follows that $\forall G_s \in \mathbf{G}_s, G_s \subseteq G_{\text{inv}}$. Otherwise if $G_s$ contains subgraphs of $G_{env}$, $G_{s1}$, or $G_{s2}$ (Figure 1(b)), the different $G_{env}$ caused by $\epsilon, \epsilon'$ and the fluctuations in $S_1, S_2$ will lead to $I(f(G_s), f(G'_s)) < 1$ as discussed above. Therefore, $I(f(G_s), f(G'_s)) = 1 \Rightarrow G_s \subseteq G_{\text{inv}}$. In addition, according to (1), $G_{\text{inv}}$ is also included in the set $\mathbf{G}_s$ since $I(f_z(G_{\text{inv}}), f_z(G'_{\text{inv}})) = 1$, which satisfies the definition of $\mathbf{G}_s$. Therefore, the solution for both "$\forall G_s \in \mathbf{G}_s, G_s \subseteq G_{\text{inv}}$" and "$G_{\text{inv}} \in \mathbf{G}_s$" reveals that $G_{\text{inv}}$ is the largest subgraph in $\mathbf{G}_s$, i.e., $G_{\text{inv}} = \text{argmax}_{G_s \in \mathbf{G}_s} |G_s|$. □

**Theorem 5.1** *If (1)* $\exists (\mathbf{X}_1, \cdots, \mathbf{X}_j) \in P^{train}$ *from at least 2 environments, s.t.* $(\mathbf{X}_{1var}, \cdots, \mathbf{X}_{jvar})$ *span* $\mathbb{R}^j$, *and (2)* $\forall \mathbf{X}_1 \neq \mathbf{X}_2$, *the GNN encoder of* $f_\psi$ *maps* $G_1 = (\mathbf{X}_1, \mathbf{A}, \mathbf{E})$ *and* $G_2 = (\mathbf{X}_2, \mathbf{A}, \mathbf{E})$ *to different embeddings, then with* $\hat{y} = f_\psi(\mathbf{X}, \mathbf{A}, \mathbf{E})$, $\hat{y} \perp\!\!\!\perp \mathbf{X}_{var}$ *as* $n_A \to \infty$.

We theoretically prove the statements and Theorem 5.1 for FeatX. We propose to learn and apply a mask $\mathbf{M}$ and perturb the non-causal node features to achieve extrapolation w.r.t. $\mathbf{x}_{\text{var}}$, without altering the topological structure of the graph. Let the domain for $\mathbf{x}$ be denoted as $\mathcal{D}$, which is assumed to be accessible. Valid extrapolations must generate augmented samples with node feature $\mathbf{X}_A \in \mathcal{D}$ while $\mathbf{X}_A \nsim P^{\text{train}}(\mathbf{X})$. Since $\mathbf{x}$ is a vector, $\mathcal{D}$ is also a vector, in which each element gives the domain of an element in $\mathbf{x}$. We ensure the validity of extrapolation with the generalized modulo operation $\text{mod}$, which we define as

$$\mathbf{X} \bmod \mathcal{D} = \mathbf{X} + i * abs(\mathcal{D}), s.t. \mathbf{X} \bmod \mathcal{D} \in \mathcal{D}, \tag{13}$$

where $i$ is any integer and $abs(\mathcal{D})$ calculates the range length of $\mathcal{D}$. Therefore, $\forall \mathbf{X} \in \mathbb{R}^p, (\mathbf{X} \bmod \mathcal{D}) \in \mathcal{D}$. Given each pair of samples $\mathbf{D}_{\varepsilon_1}, \mathbf{D}_{\varepsilon_2}$ with the same label $y$ but different environments $\varepsilon_1$ and $\varepsilon_2$, FeatX produces

$$\mathbf{X}_A = \mathbf{M} \times ((1 + \lambda)\mathbf{X}_{\varepsilon_1} - \lambda'\mathbf{x}_{\varepsilon_2}) \bmod \mathcal{D} + \overline{\mathbf{M}} \times \mathbf{X}_{\varepsilon_1},$$
$$(\mathbf{A}, \mathbf{E}) = (\mathbf{A}_{\varepsilon_1}, \mathbf{E}_{\varepsilon_1}),$$

where $\lambda, \lambda' \sim \mathcal{N}(a, b)$ is sampled for each data pair. During the process, the augmented samples form a new environment.

We prove Theorem 5.1, showing that, under certain conditions, FeatX substantially solves feature shifts on the selected variant features for node-level tasks. The proof also evidences that our extrapolation spans the feature space outside $P^{\text{train}}(\mathbf{X})$ for $\mathbf{x}_{\text{var}}$, transforming OOD areas to ID. Let $n_A$ be the number of samples FeatX generates and $f_\psi$ be the well-trained network with FeatX applied.

*Proof.* Condition is given that

$$\exists (\mathbf{X}_1, \cdots, \mathbf{X}_j) \in P^{\text{train}}, (\mathbf{X}_{1\text{var}}, \cdots, \mathbf{X}_{j\text{var}})\text{span}\mathbb{R}^j. \tag{14}$$

Therefore, by definition,

$$\forall \mathbf{u} \in \mathbb{R}^j, \exists \mathbf{t} = (t_1, t_2, \cdots, t_j), t_1, t_2, \cdots, t_j \in \mathbb{R}^j, s.t. \mathbf{u} = t_1 \mathbf{X}_{1\text{var}} + \cdots + t_j \mathbf{X}_{j\text{var}}. \tag{15}$$

The operation to generate $\mathbf{X}_A$ gives

$$\mathbf{X}_A = \mathbf{M} \times ((1 + \lambda)\mathbf{X}_{\varepsilon_1} - \lambda'\mathbf{X}_{\varepsilon_2}) \bmod \mathcal{D} + \overline{\mathbf{M}} \times \mathbf{X}_{\varepsilon_1}, \tag{16}$$

so we have

$$\mathbf{X}_{A\text{var}} = ((1 + \lambda)\mathbf{X}_{\varepsilon_1\text{var}} - \lambda'\mathbf{X}_{\varepsilon_2\text{var}}) \bmod \mathcal{D}. \tag{17}$$

For $\forall \mathbf{u}, \exists \mathbf{t} = (t_1, t_2, \cdots, t_j)$ and $(\mathbf{X}_1, \cdots, \mathbf{X}_j) \in P^{\text{train}}$ from at least 2 environments. Without loss of generality, we assume that $\mathbf{X}_1$ and $\mathbf{X}_2$ are from different environments $\varepsilon_1$ and $\varepsilon_2$. With $n_A \to \infty$, there will exist an augmentation sampled between $\mathbf{X}_1$ and $\mathbf{X}_2$, and since $\lambda \sim \mathcal{N}(a, b), \lambda \in \mathbb{R}$,

$$\exists \mathbf{X}_A^1 \quad s.t. \mathbf{X}_{A\text{var}}^1 = ((1 + \lambda)\mathbf{X}_{1\text{var}} - \lambda'\mathbf{X}_{2\text{var}}) \bmod \mathcal{D}$$
$$= (1 + \lambda)\mathbf{X}_{1\text{var}} - \lambda'\mathbf{X}_{2\text{var}} + n_1 * abs(\mathcal{D}), 1 + \lambda = t_1, -\lambda' = t_2,$$

where $n_1$ is an integer. Equivalently,

$$X^1_{A\,var} = t_1 X_{1\,var} + t_2 X_{2\,var} + n_1 * abs(\mathcal{D}). \tag{18}$$

The augmentation sample $X^1_A$ belongs to a new environment, thus in a different environment from $X_1, \cdots, X_j$. Similarly, with $n_A \to \infty$, there will exist an augmentation sampled between $X^1_A$ and $X_3$,

$$\exists X^2_A \quad s.t. X^2_{A\,var} = ((1+\lambda)X^1_{A\,var} - \lambda' X_{3\,var}) \bmod \mathcal{D}$$
$$= (1+\lambda)X^1_{A\,var} - \lambda' X_{3\,var} + n_2 * abs(\mathcal{D}), \lambda = 0, -\lambda' = t_3$$

where $n_2$ is an integer. Equivalently,

$$X^2_{A\,var} = X^1_{A\,var} + t_3 X_{3\,var} + n_2 * abs(\mathcal{D}) = t_1 X_{1\,var} + t_2 X_{2\,var} + t_3 X_{3\,var} + (n_1 + n_2) * abs(\mathcal{D}). \tag{19}$$

The augmentation sample $X^2_A$ also belongs to the new environment.

Recursively, with $n_A \to \infty$, there will exist an augmentation

$$\exists X^{j-1}_A \quad s.t. X^{j-1}_{A\,var} = t_1 X_{1\,var} + t_2 X_{2\,var} + \cdots + t_j X_{j\,var} + (n_1 + n_2 + \cdots + n_{j-1}) * abs(\mathcal{D}). \tag{20}$$

Since for $\boldsymbol{u}$, we have $\boldsymbol{u} = t_1 X_{1\,var} + \cdots + t_j X_{j\,var}$, therefore, $X^{j-1}_{A\,var} = \boldsymbol{u} + (n_1 + n_2 + \cdots + n_{j-1}) * abs(\mathcal{D})$. With $\boldsymbol{u} \in \mathbb{R}^j$ and $X^{j-1}_{A\,var} = ((1+\lambda)X^{j-2}_{A\,var} - \lambda' X_{j\,var}) \bmod \mathcal{D} \in \mathbb{R}^j$ by the definition of $\bmod \mathcal{D}$, we have

$$(n_1 + n_2 + \cdots + n_{j-1}) * abs(\mathcal{D}) = 0 \text{ and } X^{j-1}_{A\,var} = \boldsymbol{u}. \tag{21}$$

Therefore, we prove that with $n_A \to \infty$,

$$\forall \boldsymbol{u} \in \mathbb{R}^j, \text{ there exists an augmentation sample } X^{j-1}_A \quad s.t. X^{j-1}_{A\,var} = \boldsymbol{u}. \tag{22}$$

That is, the extrapolation strategy of FeatX spans the feature space for $\boldsymbol{x}_{var}$.

With the above result, for the feature space of $\boldsymbol{x}_{var}$, every data point is reachable. As $n_A \to \infty$, every data point of $\boldsymbol{x}_{var}$ is reached at least once. Let a group of samples with selected and preserved causal features $X_{inv*}$ be $M \times X_{var} + \overline{M} \times X_{inv*}$, where $X_{var} \leftarrow \forall \boldsymbol{x}_{var} \in \mathbb{R}^j$. Since $\forall X_1 \neq X_2$, the GNN encoder maps $G_1 = (X_1, A, E)$ and $G_2 = (X_2, A, E)$ to different embeddings, all different samples from $M \times X_{var} + \overline{M} \times X_{inv*}$ are encoded into different embeddings, while all having the same label $y$. For the well-trained network $f_\psi$, the group of embeddings $Z | (M \times X_{var} + \overline{M} \times X_{inv*})$ are all predicted into class $\hat{y} = y$. In this case,

$$\forall X_{var} \in \mathbb{R}^j, \quad \hat{y} = f_\psi(M \times X_{var} + \overline{M} \times X_{inv*}, A, E) = y, \tag{23}$$

therefore

$$\hat{y} \perp\!\!\!\perp X_{var} \quad \text{as} \quad n_A \to \infty. \tag{24}$$

This completes the proof. $\qquad\square$

Theorem 5.1 states that, given sufficient diversity in environment information and expressiveness of GNN, FeatX can achieve invariant prediction regarding the selected variant features. Therefore, FeatX possesses the capability to generalize over distribution shifts on the selected variant features. Extending on the accuracy of non-causal selection, if $\boldsymbol{x}_{var*} = \boldsymbol{x}_{var}$, we achieve causally-invariant prediction in feature-based OOD tasks. Thus, FeatX possesses the potential to solve feature distribution shifts.

# H  EXPERIMENTAL DETAILS

We further describe experimental details in the following sections.

## H.1 DATASET DETAILS

To evidence the generalization improvements of structure extrapolation, we evaluate G-Splice on 8 graph-level OOD datasets with structure shifts. We adopt 5 datasets from the GOOD benchmark (Gui et al., 2022a), GOODHIV-size, GOODHIV-scaffold, GOODSST2-length, GOODMotif-size, and GOODMotif-base, using the covariate shift split from GOOD. GOOD-HIV is a real-world molecular dataset with shift domains scaffold and size. The first one is Bemis-Murcko scaffold (Bemis and Murcko, 1996) which is the two-dimensional structural base of a molecule. The second one is the number of nodes in a molecular graph. GOOD-SST2 is a real-world natural language sentimental analysis dataset with sentence lengths as domain, which is equivalent to the graph size. GOOD-Motif is a synthetic dataset specifically designed for structure shifts. Each graph is generated by connecting a base graph and a motif, with the label determined by the motif solely. The shift domains are the base graph type and the graph size. We construct another natural language dataset Twitter (Yuan et al., 2020) following the OOD splitting process of GOOD, with length as the shift domain. In addition, we adopt protein dataset DD and molecular dataset NCI1 following Bevilacqua et al. (2021), both with size as the shift domain. All datasets possess structure shifts as we have discussed, thus proper benchmarks for structural OOD generalization.

To show the OOD the generalization improvements of feature extrapolation, we evaluate FeatX on 5 graph OOD datasets with feature shifts. We adopt 5 datasets of the covariate shift split from the GOOD benchmark. GOOD-CMNIST is a semi-artificial dataset designed for node feature shifts. It contains image-transformed graphs with color features manually applied, thus the shift domain color is structure-irrelevant. The other 4 datasets are node-level. GOOD-Cora is a citation network dataset with "word" shift, referring to the word diversity feature of a node. The input is a small-scale citation network graph, in which nodes represent scientific publications and edges are citation links. The shift domain is word, the word diversity defined by the selected-word-count of a publication. GOOD-Twitch is a gamer network dataset, with the node feature "language" as shift domain. The nodes represent gamers and the edge represents the friendship connection of gamers. The binary classification task is to predict whether a user streams mature content. The shift domain of GOOD-Twitch is user language. GOOD-WebKB is a university webpage network dataset. A node in the network represents a webpage, with words appearing in the webpage as node features. Its 5-class prediction task is to predict the owner occupation of webpages, and the shift domain is university, which is implied in the node features. GOOD-CBAS is a synthetic dataset. The input is a graph created by attaching 80 house-like motifs to a 300-node Barabási–Albert base graph, and the task is to predict the role of nodes. It includes colored features as in GOOD-CMNIST so that OOD algorithms need to tackle node color differences, which is also typical as feature shift. All shift domains are structure-irrelevant and provide specific evaluation for feature extrapolation.

Following prior works (Wu et al., 2022b; Gui et al., 2022a), we also create another synthetic dataset FSMotif. The GOOD benchmark we use for major evaluation in the paper does not contain OOD datasets with shifts on both structure and feature, which cannot provide evaluation for the combined effectiveness of FLE and SLE. We create FSMotif, with complex shifts on both structure and feature, to prove the superiority of our methods when used concurrently. FSMotif is a synthetic dataset where each graph is generated by connecting a base graph and a motif, with the label determined by the motif solely and all nodes given color features. The shift domains are 1.the base graph type and the color feature, and 2.the graph size and the color feature. Specifically, we generate graphs using seven colors, five label irrelevant base graphs (wheel, tree, ladder, star, and path), and three label determining motifs (house, cycle, and crane).

## H.2 SETUP DETAILS

We conduct experiments on 8 datasets with 16 baseline methods to evaluate G-Splice, and on 5 datasets with 16 baselines for FeatX. As a common evaluation protocol, datasets for OOD tasks provides OOD validation/test sets (Gui et al., 2022a; Bevilacqua et al., 2021) to evaluate the model's OOD generalization abilities. Some datasets also provide ID validation/test sets for comparison (Gui et al., 2022a). For all experiments, we select the best checkpoints for OOD tests according to results on OOD validation sets; ID validation and ID test are also used for comparison if available. For graph prediction and node prediction tasks, we respectively select strong and commonly acknowledged GNN backbones. For each dataset, we use the same GNN backbone for all baseline methods for fair comparison. For graph prediction tasks, we use GIN-Virtual Node (Xu et al., 2019a; Gilmer et al.,

2017) as the GNN backbone. As an exception, for GOOD-Motif we adopt GIN (Xu et al., 2019a) as the GNN backbone, since we observe from experiments that the global information provided by virtual nodes would interrupt the training process here. For node prediction tasks, we adopt GraphSAINT (Zeng et al., 2020) and use GCN (Kipf and Welling, 2017) as the GNN backbone. For all the experiments, we use the Adam optimizer, with a weight decay tuned from the set {0, 1e-2, 1e-3, 1e-4} and a dropout rate of 0.5. The number of convolutional layers in GNN models for each dataset is tuned from the set {3, 5}. We use mean global pooling and the RELU activation function, and the dimension of the hidden layer is 300. We select the maximum number of epochs from {100, 200, 500}, the initial learning rate from {1e-3, 3e-3, 5e-3, 1e-4}, and the batch size from {32, 64, 128} for graph-level and {1024, 4096} for node-level tasks. All models are trained to converge in the training process. For computation, we generally use one NVIDIA GeForce RTX 2080 Ti for each single experiment.

### H.3 Hyperparameter Selection

In all experiments, we perform hyperparameter search to obtain experimental results that can well-reflect the performance potential of models. For each dataset and method, we search from a hyperparameter set and select the optimal one based on OOD validation metric scores.

For each baseline method, we tune one or two algorithm-specific hyperparameters. For IRM and Deep Coral, we tune the weight for penalty loss from {1e-1, 1, 1e1, 1e2} and {1, 1e-1, 1e-2, 1e-3}, respectively. For VREx, we tune the weight for VREx's loss variance penalty from {1, 1e1, 1e2, 1e3}. For GroupDRO, we tune the step size from {1e-1, 1e-2, 1e-3}. For DANN, we tune the weight for domain classification penalty loss from {1, 1e-1, 1e-2, 1e-3}. For Graph Mixup, we tune the alpha value of its Beta function from {0.4, 1, 2}. The Beta function is used to randomize the lamda weight, which is the weight for mixing two instances up. For DIR, we tune the causal ratio for selecting causal edges from {0.2, 0.4, 0.6, 0.8} and loss control from {1e1, 1, 1e-1, 1e-2}. For EERM, we tune the learning rate for reinforcement learning from {1e-2, 1e-3, 5e-3, 1e-4} and the beta value to trade off between mean and variance from {1, 2, 3}. For SRGNN, we tune the weight for shift-robust loss calculated by central moment discrepancy from {1e-4, 1e-5, 1e-6}. For DropNode, DropEdge and MaskFeature, we tune the drop/mask percentage rate from {0.05, 0.1, 0.15, 0.2, 0.3}. For FLAG, we set the number of ascending steps $M = 3$ and tune the ascent step size from the set {1e-2, 1e-3, 5e-3, 1e-4}. For LISA, we tune the parameters of the Beta function in the same way as Graph Mixup. For G-Mixup, we set the augmentation number to 10, tune the augmentation ratio from {0.1, 0.2, 0.3} and the lambda range from {[0.1,0.2], [0.2,0.3]}. For GIL, we tune IGA lambda value from {1e-2, 1e-3, 1e-4} and set top ratio of subgraphs and number of environments by its originally reported optimum. For CIGA, we tune the size ratio of the causal subgraphs from {0.4, 0.6, 0.8}, while contrastive loss and hinge loss weights from {0.5, 1, 2}.

For G-Splice, we tune the percentage of augmentation from {0.6, 0.8, 1.0}. The actual number of component graphs $f$ is tuned from {2, 3, 4}, and the augmentation selection is tuned as a 3-digit binary code representing the 3 options, with at least one option applied. For the pre-training of the bridge generation, hyperparameters regularizing the bridge attribute and KL divergence $\alpha$ and $\beta$ are tuned from {1.5, 1, 0.5, 0.1}. When the additional VREx-like regularization is applied, we tune the weight of loss variance penalty from {1, 1e1, 1e2}. For FeatX, we tune the the shape parameter $a$ and scale parameter $b$ of the gamma function $\Gamma(a, b)$ from {2, 3, 5, 7, 9} and {0.5, 1.0, 2.0}, respectively.

## I  Ablation Studies

### I.1  Bridge Generation Studies for G-Splice

#### I.1.1  VAE as Bridge Generator

In this work, we adopt conditional VAE (Kingma and Welling, 2013; Kipf and Welling, 2016; Sohn et al., 2015) as the major bridge generator for G-Splice due to its adequate capability and high efficiency. We show empirically that VAEs are well suitable for our task.

We reconstruct the generation process with diffusion model (Ho et al., 2020), a generative model with high capability and favorable performances across multiple tasks. Diffusion models consist of a diffusion process which progressively distorts a data point to noise, and a generative denoising

process which approximates the reverse of the diffusion process. In our case, the diffusion process adds Gaussian noise independently on each node and edge features encoded into one-hot vectors at each time step. Then the denoising network is trained to predict the noises, and we minimize the error between the predicted noise and the true noise computed in closed-form. During sampling, we iteratively sample bridge indexes and attribute values, and then map them back to categorical values in order that we obtain a valid graph. We compare performances and computational efficiency of the two generative models. As a baseline for bridge generation, we also present the results of random bridges, where bridges of predicted number and corresponding attributes are randomly sampled from the group of component graphs. Note that we do not apply the regularization in these experiments.

Table 4: Comparison on bridge generation methods. G-Splice-Rand, G-Splice-VAE, and G-Splice-Diff show the performance of G-Splice on GOODHIV with random bridge, VAE generated bridge, diffusion model generated bridge, respectively. The train time ratio presents the entire training duration of a method, including module pre-training time, divided by the training duration of G-Splice-Rand in average.

| Method | GOOD-HIV-size↑ | | GOOD-HIV-scaffold↑ | | Train time ratio |
|---|---|---|---|---|---|
| | $ID_{ID}$ | $OOD_{OOD}$ | $ID_{ID}$ | $OOD_{OOD}$ | |
| ERM | 83.72±1.06 | 59.94±2.86 | 82.79±1.10 | 69.58±1.99 | 0.57 |
| G-Splice-Rand | 83.25±0.96 | 62.36±2.25 | **84.33**±0.69 | 71.89±2.80 | 1 |
| G-Splice-VAE | **84.75**±0.18 | **64.46**±1.38 | 83.23±0.97 | 72.82±1.16 | 1.52 |
| G-Splice-Diff | 84.35±0.35 | 64.09±0.82 | 83.45±0.97 | **72.95**±1.80 | 20.25 |

As can be observed in Table 4, OOD test results from the two generative models are comparable, both significantly improving over G-Splice-Rand. The diffusion model may be slightly limited in performance gain due to the discreteness approximations during sampling. The results implies the necessity of generative models in the splicing operation for overall structural extrapolation. Meanwhile, this shows that VAE is capable of the bridge generation task. In contract, the training duration of diffusion model is 13 times that of VAE due to the sampling processes through massive time steps. Overall, we obtain comparable performances from the two generative models, while VAEs are much less expensive computationally. Therefore, empirical results demonstrate that adopting VAE as our major bridge generator is well suitable.

### I.1.2 Bridge Generation Design

As we have introduced in Sec 4.1, we generate bridges of predicted number along with corresponding edge attributes between given component graphs to splice graphs. We do not include new nodes as a part of the bridge, since we aim at preserving the local structures of the component graphs and extrapolating certain global features. More manually add-on graph structures provide no extrapolation significance, while their interpolation influence are not proven beneficial. We evidence the effectiveness of our design with experiments. We additionally build a module to generate nodes in the bridges. The number of nodes is predicted with a pre-trained predictive model and then a generative model generates the node features. Moreover, we evaluate the results with fixed instead of generated bridge attributes. The performances with our original bridge generator, node generation applied and edge attribute generation removed is summarized as follows. Note that we do not apply the regularization in these experiments.

Table 5: Comparison of bridge generation designs. G-Splice orig, G-Splice + node, and G-Splice - attr show the performance of G-Splice on GOODHIV with the original bridge generator, node generation applied and edge attribute generation removed, respectively.

| Method | GOOD-HIV-size↑ | | GOOD-HIV-scaffold↑ | |
|---|---|---|---|---|
| | $ID_{ID}$ | $OOD_{OOD}$ | $ID_{ID}$ | $OOD_{OOD}$ |
| ERM | 83.72±1.06 | 59.94±2.86 | 82.79±1.10 | 69.58±1.99 |
| G-Splice orig | **84.75**±0.18 | **64.46**±1.38 | 83.23±0.97 | **72.82**±1.16 |
| G-Splice + node | 83.14±0.82 | 62.65±2.67 | **84.67**±0.48 | 71.76±1.76 |
| G-Splice - attr | 84.50±0.44 | 64.13±0.62 | 83.41±1.10 | 72.07±1.52 |

As can be observed in Table 5, OOD test performances from the original bridge generator remains the highest. Without attribute generation, fixed bridge attributes degrades the overall performance due to the manual feature of the bridges, which may mislead the model with spurious information. When we include nodes as a part of the bridge, similarly the manually add-on graph structure may inject spurious information to the model and perturb the preservation of local structures, leading to limited improvements. This evidences the effectiveness of our design for bridge generation.

## I.2 COMPARISON OF EXTRAPOLATION PROCEDURES FOR G-SPLICE

We evidence that certain extrapolation procedures specifically benefit size or base/scaffold shifts, as our theoretical analysis in Sec. 4 and 5. For size and base/scaffold shifts on GOODHIV and GOODMotif, we extrapolation with each of the three augmentation options, $G_{inv}$, $G_{inv} + f \cdot G_{env}$ and $f \cdot G$, individually and together, and compare the OOD performances. Note that we apply the VREx-like regularization in these experiments.

Table 6: Comparison of extrapolation procedures for G-Splice. Performances of G-Splice on GOOD-HIV and GOODMotif with augmentation options single causal subgraph, causal and environmental subgraphs spliced, whole graphs spliced, and all three options applied. Optimal show the performances with options selected after hyperparameter tuning.

| G-Splice | GOOD-HIV-size↑ | | GOOD-HIV-scaffold↑ | | GOOD-Motif-size↑ | | GOOD-Motif-base↑ | |
|---|---|---|---|---|---|---|---|---|
| | ID$_{ID}$ | OOD$_{OOD}$ | ID$_{ID}$ | OOD$_{OOD}$ | ID$_{ID}$ | OOD$_{OOD}$ | ID$_{ID}$ | OOD$_{OOD}$ |
| $G_{inv}$ | 83.90±0.40 | 63.04±2.40 | 83.19±0.69 | 72.04±0.96 | 91.10±0.10 | 76.95±4.52 | 92.12±0.14 | 69.59±3.67 |
| $G_{inv} + f \cdot G_{env}$ | **85.40**±0.82 | 62.65±2.16 | 83.97±0.57 | 72.83±1.86 | 91.08±0.63 | 72.10±5.43 | 92.01±0.23 | 73.92±5.44 |
| $f \cdot G$ | 84.75±0.18 | 63.94±1.46 | **85.10**±0.67 | 73.14±1.05 | 91.93±0.21 | **85.07**±4.50 | 92.12±0.15 | 76.19±10.99 |
| All | 85.09±0.61 | 63.16±1.38 | 83.47±0.45 | 72.39±0.52 | **92.00**±0.30 | 82.86±2.53 | 92.01±0.16 | 80.09±12.10 |
| Optimal | 84.85±0.19 | **65.56**±0.34 | 83.36±0.40 | **73.28**±0.16 | 91.93±0.21 | 85.07±4.50 | **92.14**±0.29 | **83.96**±7.38 |

Let the three augmentation options, $G_{inv}$, $G_{inv} + f \cdot G_{env}$ and $f \cdot G$ be numbered 1, 2, and 3. The optimal augmentation options for GOOD-HIV-size, GOOD-HIV-scaffold, GOOD-Motif-size, and GOOD-Motif-base after hyperparameter tuning are 1+3, 2+3, 3, and 2+3, respectively. As can be observed from Table 6, $G_{inv}$ and $f \cdot G$ have advantages in size shifts, while $G_{inv} + f \cdot G_{env}$ and $f \cdot G$ are better for base/scaffold shifts. This matches our theoretical analysis of augmentation procedures. For size distribution shifts, $G_{inv}$ and $f \cdot G$ environments enable size extrapolation by creating smaller and larger graphs outside the training distribution, respectively. For base/scaffold distribution shifts, the two new environments respectively construct graphs without base/scaffold, and graphs with $f$ base/scaffolds, achieving base extrapolation with new base/scaffolds introduced. Splicing whole graphs has the advantage of extrapolating to larger graphs, simplicity in operation, and little loss in local structural information. Extracting subgraphs allows better flexibility for G-Splice, making graphs smaller than the training size accessible. In addition, the performance gain from $f \cdot G$ shows the effectiveness of the simple splicing strategy by itself.

## I.3 ABLATION STUDIES ON FEATX

FeatX enables extrapolation w.r.t. the selected variant features. By generating causally valid samples with OOD node features, FeatX essentially expands the training distribution range. Theoretical analysis evidences that our extrapolation spans the feature space outside $P^{train}(X)$ for $x_{var}$, thereby transforming OOD areas to ID. We further show with experiments that extrapolation substantially benefits feature shifts in OOD tasks compared with interpolation, which can also improve generalization by boosting learning processes. In addition, we show that our invariance mask and variance score vectors succeed in selecting non-causal features by comparisons between perturbation on selected features and all features.

As can be observed from Table 7, whether selecting non-causal features and the choice between interpolation and extrapolation both show significant influence on generalization performances. In all three datasets, extrapolation performances exceed corresponding interpolation performances with a clear gap, demonstrating the benefits of extrapolation by generating samples in OOD area that interpolation cannot reach. In GOODWebKB, perturbing selected non-causal features achieve significant improvements over perturbing all features, regardless of interpolation and extrapolation. This

Table 7: Performances w/o feature selection and extrapolation for FeatX. We show the performance comparisons of interpolating or extrapolating the selected or all features on GOODCMNIST, GOOD-WebKB and GOODCBAS.

| Feature | Perturbation | GOOD-CMNIST-color↑ | | GOOD-WebKB-university↑ | | GOOD-CBAS-color↑ | |
|---|---|---|---|---|---|---|---|
| | | $ID_{ID}$ | $OOD_{OOD}$ | $ID_{ID}$ | $OOD_{OOD}$ | $ID_{ID}$ | $OOD_{OOD}$ |
| | ERM | 77.96±0.34 | 28.60±2.01 | 38.25±0.68 | 14.29±3.24 | 89.29±3.16 | 76.00±3.00 |
| All | Interpolation | 76.62±0.46 | 29.61±2.54 | 37.56±3.67 | 15.59±2.48 | 92.00±0.15 | 81.76±1.75 |
| All | Extrapolation | 75.14±0.38 | 54.13±5.08 | 38.26±5.18 | 17.10±3.45 | **93.25**±0.43 | 84.28±3.67 |
| Selected | Interpolation | **78.15**±0.30 | 31.65±5.02 | 43.15±1.42 | 26.16±2.50 | 90.10±2.14 | 80.37±1.35 |
| Selected | Extrapolation | 69.54±1.51 | **62.49**±2.12 | **50.82**±0.00 | **32.54**±8.98 | 92.86±1.17 | **87.62**±2.43 |

evidences the effectiveness of non-causal feature selection using variance score vectors, empirically supporting our design. In GOODCMNIST and GOODCBAS, since the features are manually added colors, the effect of feature selection is not as obvious as in GOODWebKB, the real world dataset. Experimental results evidence the effectiveness of the strategies designed in FeatX.

## J METRIC SCORE AND LOSS CURVES

We report the metric score curves and loss curves for part of the datasets in Figure 2-5. As can be observed from each pair of curves, our proposed methods, G-Splice and FeatX, consistently achieve better metric scores and lower loss compared with other baselines during the learning process. This evidences the substantial improvements achieved by structure and feature extrapolation, which benefits OOD generalization in essence.

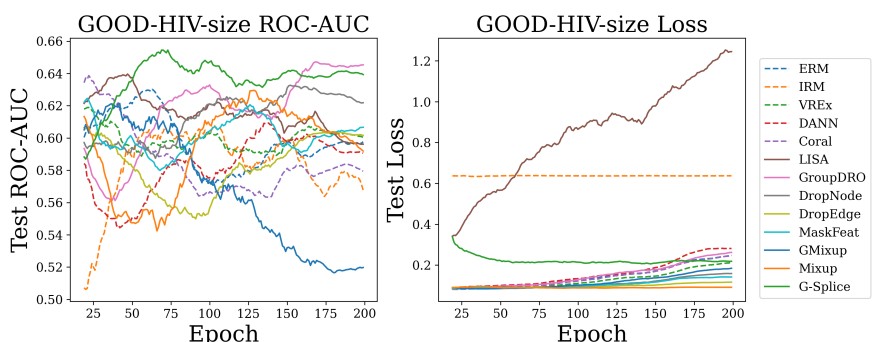

Figure 2: ROC-AUC score curve and loss curve for GOODHIV-size.

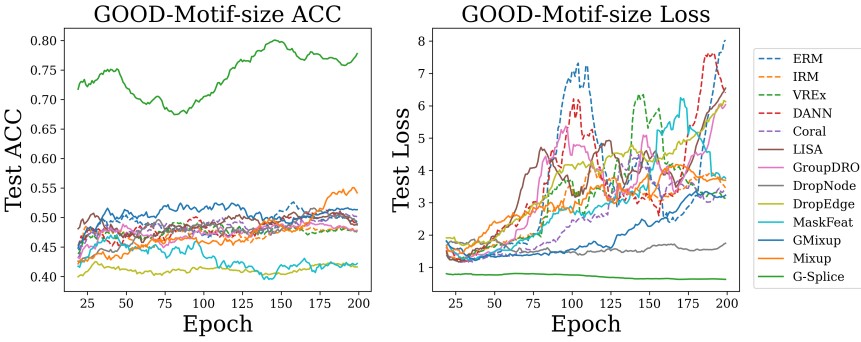

Figure 3: Accuracy score curve and loss curve for GOODMotif-size.

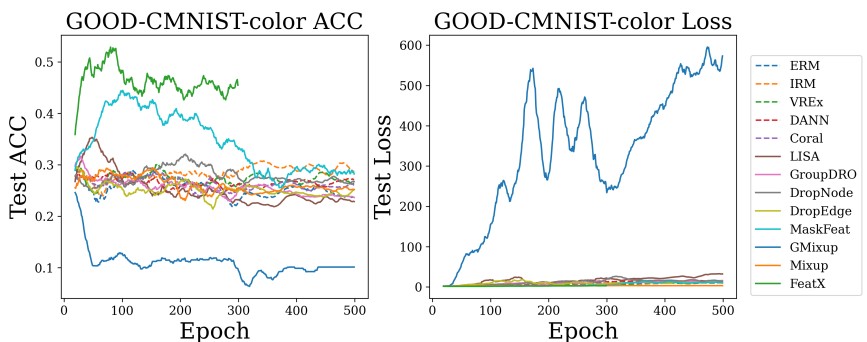

Figure 4: Accuracy score curve and loss curve for GOODCMNIST-color.

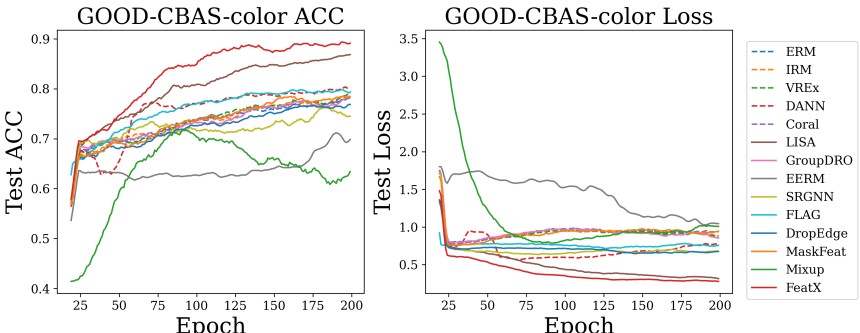

Figure 5: Accuracy score curve and loss curve for GOODCBAS-color.

