so we have

$$\boldsymbol{X}_{A\,\text{var}} = ((1+\lambda)\boldsymbol{X}_{\varepsilon_1\,\text{var}} - \lambda'\boldsymbol{X}_{\varepsilon_2\,\text{var}}) \bmod \mathcal{D}. \tag{17}$$

For $\forall \boldsymbol{u}$, $\exists \boldsymbol{t} = (t_1, t_2, \cdots, t_j)$ and $(\boldsymbol{X}_1, \cdots, \boldsymbol{X}_j) \in P^{\text{train}}$ from at least 2 environments. Without loss of generality, we assume that $\boldsymbol{X}_1$ and $\boldsymbol{X}_2$ are from different environments $\varepsilon_1$ and $\varepsilon_2$. With $n_A \to \infty$, there will exist an augmentation sampled between $\boldsymbol{X}_1$ and $\boldsymbol{X}_2$, and since $\lambda \sim \mathcal{N}(a, b), \lambda \in \mathbb{R}$,

$$\exists \boldsymbol{X}_A^1 \quad s.t. \boldsymbol{X}_{A\,\text{var}}^1 = ((1+\lambda)\boldsymbol{X}_{1\,\text{var}} - \lambda'\boldsymbol{X}_{2\,\text{var}}) \bmod \mathcal{D}$$
$$= (1+\lambda)\boldsymbol{X}_{1\,\text{var}} - \lambda'\boldsymbol{X}_{2\,\text{var}} + n_1 * abs(\mathcal{D}), 1+\lambda = t_1, -\lambda' = t_2,$$

where $n_1$ is an integer. Equivalently,

$$\boldsymbol{X}_{A\,\text{var}}^1 = t_1 \boldsymbol{X}_{1\,\text{var}} + t_2 \boldsymbol{X}_{2\,\text{var}} + n_1 * abs(\mathcal{D}). \tag{18}$$

The augmentation sample $\boldsymbol{X}_A^1$ belongs to a new environment, thus in a different environment from $\boldsymbol{X}_1, \cdots, \boldsymbol{X}_j$. Similarly, with $n_A \to \infty$, there will exist an augmentation sampled between $\boldsymbol{X}_A^1$ and $\boldsymbol{X}_3$,

$$\exists \boldsymbol{X}_A^2 \quad s.t. \boldsymbol{X}_{A\,\text{var}}^2 = ((1+\lambda)\boldsymbol{X}_{A\,\text{var}}^1 - \lambda'\boldsymbol{X}_{3\,\text{var}}) \bmod \mathcal{D}$$
$$= (1+\lambda)\boldsymbol{X}_{A\,\text{var}}^1 - \lambda'\boldsymbol{X}_{3\,\text{var}} + n_2 * abs(\mathcal{D}), \lambda = 0, -\lambda' = t_3$$

where $n_2$ is an integer. Equivalently,

$$\boldsymbol{X}_{A\,\text{var}}^2 = \boldsymbol{X}_{A\,\text{var}}^1 + t_3 \boldsymbol{X}_{3\,\text{var}} + n_2 * abs(\mathcal{D}) = t_1 \boldsymbol{X}_{1\,\text{var}} + t_2 \boldsymbol{X}_{2\,\text{var}} + t_3 \boldsymbol{X}_{3\,\text{var}} + (n_1 + n_2) * abs(\mathcal{D}). \tag{19}$$

The augmentation sample $\boldsymbol{X}_A^2$ also belongs to the new environment.

Recursively, with $n_A \to \infty$, there will exist an augmentation

$$\exists \boldsymbol{X}_A^{j-1} \quad s.t. \boldsymbol{X}_A^{j-1}{}_{\text{var}} = t_1 \boldsymbol{X}_{1\,\text{var}} + t_2 \boldsymbol{X}_{2\,\text{var}} + \cdots + t_j \boldsymbol{X}_{j\,\text{var}} + (n_1 + n_2 + \cdots + n_{j-1}) * abs(\mathcal{D}). \tag{20}$$

Since for $\boldsymbol{u}$, we have $\boldsymbol{u} = t_1 \boldsymbol{X}_{1\,\text{var}} + \cdots + t_j \boldsymbol{X}_{j\,\text{var}}$, therefore, $\boldsymbol{X}_A^{j-1}{}_{\text{var}} = \boldsymbol{u} + (n_1 + n_2 + \cdots + n_{j-1}) * abs(\mathcal{D})$. With $\boldsymbol{u} \in \mathbb{R}^j$ and $\boldsymbol{X}_A^{j-1}{}_{\text{var}} = ((1+\lambda)\boldsymbol{X}_A^{j-2}{}_{\text{var}} - \lambda'\boldsymbol{X}_{j\,\text{var}}) \bmod \mathcal{D} \in \mathbb{R}^j$ by the definition of $\bmod \mathcal{D}$, we have

$$(n_1 + n_2 + \cdots + n_{j-1}) * abs(\mathcal{D}) = 0 \text{ and } \boldsymbol{X}_A^{j-1}{}_{\text{var}} = \boldsymbol{u}. \tag{21}$$

Therefore, we prove that with $n_A \to \infty$,

$$\forall \boldsymbol{u} \in \mathbb{R}^j, \text{ there exists an augmentation sample } \boldsymbol{X}_A^{j-1} \quad s.t. \boldsymbol{X}_A^{j-1}{}_{\text{var}} = \boldsymbol{u}. \tag{22}$$

That is, the extrapolation strategy of FeatX spans the feature space for $\boldsymbol{x}_{\text{var}}$.

With the above result, for the feature space of $\boldsymbol{x}_{\text{var}}$, every data point is reachable. As $n_A \to \infty$, every data point of $\boldsymbol{x}_{\text{var}}$ is reached at least once. Let a group of samples with selected and preserved causal features $\boldsymbol{X}_{\text{inv}*}$ be $\boldsymbol{M} \times \boldsymbol{X}_{\text{var}} + \overline{\boldsymbol{M}} \times \boldsymbol{X}_{\text{inv}*}$, where $\boldsymbol{X}_{\text{var}} \leftarrow \forall \boldsymbol{x}_{\text{var}} \in \mathbb{R}^j$. Since $\forall \boldsymbol{X}_1 \neq \boldsymbol{X}_2$, the GNN encoder maps $G_1 = (\boldsymbol{X}_1, \boldsymbol{A}, \boldsymbol{E})$ and $G_2 = (\boldsymbol{X}_2, \boldsymbol{A}, \boldsymbol{E})$ to different embeddings, all different samples from $\boldsymbol{M} \times \boldsymbol{X}_{\text{var}} + \overline{\boldsymbol{M}} \times \boldsymbol{X}_{\text{inv}*}$ are encoded into different embeddings, while all having the same label $y$. For the well-trained network $f_\psi$, the group of embeddings $\boldsymbol{Z}|(\boldsymbol{M} \times \boldsymbol{X}_{\text{var}} + \overline{\boldsymbol{M}} \times \boldsymbol{X}_{\text{inv}*})$ are all predicted into class $\hat{y} = y$. In this case,

$$\forall \boldsymbol{X}_{\text{var}} \in \mathbb{R}^j, \quad \hat{y} = f_\psi(\boldsymbol{M} \times \boldsymbol{X}_{\text{var}} + \overline{\boldsymbol{M}} \times \boldsymbol{X}_{\text{inv}*}, \boldsymbol{A}, \boldsymbol{