# OpenReview forum: "Graph Structure and Feature Extrapolation for Out-of-Distribution Generalization"
_ICLR.cc/2024/Conference — Submitted to ICLR 2024_

### Official Review · Reviewer_wvB3 · 2023-10-15

**Soundness:** 3 good
**Presentation:** 3 good
**Contribution:** 3 good
**Rating:** 6
**Confidence:** 2

**Summary:**

This paper proposes to use linear extrapolation to generate OOD graph data, which helps improve graph OOD generalization. This paper also provides theoretical guarantees for the proposed augmentations.

**Strengths:**

- The paper provides a linear extrapolation framework for graph data, with analyses and guarantees. (I do not read the proofs since I am not familiar with graph learning, so I can not promise that the theoretical results are right)
- The proposed GDA method shows strong empirical performance and outperforms the baselines a lot (I am not familiar with graph OOD algorithms, so I am not sure whether this method outperforms the existing SOTA)

**Weaknesses:**

- It seems that the up-to-date baseline used in this paper is in 2022, are there any new baselines in 2023?

**Questions:**

I have no other questions. I am not very familiar with graph learning, and I can not find obvious errors in this paper, so I give a score of 6. However,  I wish the AC consider my suggestions as little as possible.

---

### Official Review · Reviewer_hu8t · 2023-10-26

**Soundness:** 2 fair
**Presentation:** 2 fair
**Contribution:** 2 fair
**Rating:** 3
**Confidence:** 4

**Summary:**

This paper proposes structure and feature extrapolation for OOD generalization, achieving substantial improvement on various graph OOD datasets.

**Strengths:**

1. The proposed method is evaluated on a wide range of datasets and significant improvement is achieved on OOD data.
2. The proposed graph extrapolation is interesting.

**Weaknesses:**

1. This paper lacks qualitative or quantitative evaluation of extrapolated graphs (e.g. visualization on GOOD-CMNIST-color) and how the quality of augmented graphs affects the OOD generalization performance.
2. The quality of graph extrapolation depends on the quality of subgraph extraction, which might not be very reliable. Experimental analysis should be provided on the robustness of the method, especially when the causal/environment subgraphs are hard to find.

**Questions:**

Can you provide more results on the quality of causal/environment subgraph generation and some visualization of the result of G-Splice?

---

### Official Review · Reviewer_cpdN · 2023-11-01

**Soundness:** 3 good
**Presentation:** 2 fair
**Contribution:** 2 fair
**Rating:** 5
**Confidence:** 4

**Summary:**

This paper introduces a novel approach to tackle the challenge of out-of-distribution (OOD) generalization in the context of graph classification tasks. The primary objective of this method is to address domain adaptation issues by employing a data-centric approach that involves generating OOD data samples through the technique of data extrapolation. To this end, an environment-aware framework is proposed, which incorporates linear extrapolation techniques in both the graph's structural and feature spaces. The theoretical underpinnings of this work provide justifications for the causal validity of the generated samples obtained through linear extrapolation, ensuring their tailored nature for specific OOD shifts. Through comprehensive empirical analyses and extensive experimentation, the effectiveness of the proposed method is demonstrated, surpassing the performance of existing OOD learning and data augmentation approaches in the realm of graph tasks. The key contribution lies in the innovative design of non-Euclidean-space linear extrapolation, which facilitates the augmentation of both the graph's structural and feature spaces, thereby enabling the generation of OOD samples customized for specific shifts without compromising the inherent causal mechanisms. The obtained results consistently showcase notable improvements across diverse graph OOD tasks.

**Strengths:**

1)	The paper presents a data augmentation technique to address the OOD generalization challenge in graph-related tasks. The proposed method introduces the innovative concept of employing linear extrapolation for generating novel training samples, contributing to the advancement of the field.
2)	The paper introduces extrapolation techniques at both the structural and feature levels, providing valuable insights for addressing out-of-distribution (OOD) challenges in graph and node-level problems.
3)	The paper offers a comprehensive and clear theoretical proof, which serves as a solid foundation and provides theoretical substantiation for the proposed method in the paper.

**Weaknesses:**

1)	Insufficient experiments: In Chapter 6.2, concerning the feature shift node classification experiment, it is worth noting that there is a noticeable absence of experimental results on the GOOD-Arxiv dataset, which is characterized by a substantial volume of data. In addition, graph OOD problems include not only covariate shift but also concept shift. GOOD benchmark also provides relevant settings. It would be beneficial if you could provide results and discussions related to the data from this specific setting.
2)	The feature extrapolation (FeatX) seems unrelated to Structure Extrapolation. However, in graph structure, feature information and structural information are interrelated. Could you provide a more detailed explanation of feature extrapolation and structural extrapolation?

**Questions:**

1)	The article proposes a separate and independent data augmentation approach that extrapolates at the structural and feature levels. However, in practical scenarios, it is possible for both structural and feature offsets to occur simultaneously. The question arises as to whether this separate augmentation strategy can effectively address such a complex out-of-distribution (OOD) problem.

---

### Official Review · Reviewer_EXsh · 2023-11-01

**Soundness:** 3 good
**Presentation:** 3 good
**Contribution:** 3 good
**Rating:** 5
**Confidence:** 4

**Summary:**

The paper proposes a graph data augmentation method to tackle the problem of graph OOD generalization. The proposed augmentation strategy extrapolates both structure and feature spaces to generate OOD graph data without corrupting underlying causal mechanisms. Theoretical analysis and empirical results evidence the effectiveness of our method in solving target shifts, showing substantial and constant improvements across various graph OOD tasks.

**Strengths:**

* The paper tackles an important problem of graph OOD generalization.

* The contribution is novel and interesting.

* Paper writing is careful and clear.

* Experiments are conducted thoroughly.

**Weaknesses:**

* Table 1 is too dense.

**Questions:**

1. About non-causal feature selection: Is the variance score really measuring the causal relationship between feature elements and target, or just the correlation between them?

2. There is a line of work in cognitive science, namely the Structure Mapping Theory [1], stating that human’s generalization capability is based on the ability to transfer structural information. Recently this idea has been successfully incorporated (to some extent) in neural networks [2].  However, I do not see a clear relationship between causality and OOD generalization. I would appreciate if the authors could provide a more careful explanation on why preserving underlying causal mechanisms can help in OOD scenarios.

3. From Table 2, it seems that G-Splice plays more significant role than FeatX in improving the model’s performances. Do the authors have any explanation or analysis for this?

4. Variances of the proposed method are much higher on FSMotif dataset. Why is it?

I would be happy to increase my score if the authors could tackle my concerns.

References

[1] Gentner, Dedre. "Structure-mapping: A theoretical framework for analogy." Cognitive science 7.2 (1983): 155-170.

[2] Pham, Kha, et al. "Improving Out-of-distribution Generalization with Indirection Representations." The Eleventh International Conference on Learning Representations. 2022.

---

### Meta-Review · Area_Chair_29vY · 2023-12-15

**Metareview:**

This paper presents data augmentations for graph-structured data and uses it to achieve OOD generalization under distribution shifts, all while respecting the graph structure and implied causal relationships.

The strengths of this paper is introducing a theoretically-motivated, but practical data augmentation method for graph-structured data. This is followed by an extensive empirical analysis highlighting the efficacy of the proposed method, especially in comparison to various common baselines.

A major weakness of this paper is its content density. Furthermore, reviewers requested additional empirical results with other types of shifts and additional related work comparison. Overall, the paper requires multiple adaptations to address reviewer concerns and additional results.

**Justification For Why Not Higher Score:**

There were multiple unaddressed reviewer concerns, and the authors have not engaged in the discussion period.

**Justification For Why Not Lower Score:**

There is none

---

### Decision · Program_Chairs · 2024-01-16

Reject